# An alternative CTCF isoform antagonizes canonical CTCF occupancy and changes chromatin architecture to promote apoptosis

Jiao Li [1,2,3,4], Kaimeng Huang [1,2,3], Gongcheng Hu[1,2,3,4], Isaac A. Babarinde[5], Yaoyi Li[1,2,3,4], Xiaotao Dong[1,2,3,4], Yu-Sheng Chen[6], Liping Shang[1], Wenjing Guo[1], Junwei Wang[1], Zhaoming Chen[1,2], Andrew P. Hutchins [5], Yun-Gui Yang [4,6] & Hongjie Yao [1,2,3,4]

CTCF plays key roles in gene regulation, chromatin insulation, imprinting, X chromosome inactivation and organizing the higher-order chromatin architecture of mammalian genomes. Previous studies have mainly focused on the roles of the canonical CTCF isoform. Here, we explore the functions of an alternatively spliced human CTCF isoform in which exons 3 and 4 are skipped, producing a shorter isoform (CTCF-s). Functionally, we find that CTCF-s competes with the genome binding of canonical CTCF and binds a similar DNA sequence. CTCF-s binding disrupts CTCF/cohesin binding, alters CTCF-mediated chromatin looping and promotes the activation of IFI6 that leads to apoptosis. This effect is caused by an abnormal long-range interaction at the IFI6 enhancer and promoter. Taken together, this study reveals a non-canonical function for CTCF-s that antagonizes the genomic binding of canonical CTCF and cohesin, and that modulates chromatin looping and causes apoptosis by stimulating IFI6 expression.

[1] CAS Key Laboratory of Regenerative Biology, Joint School of Life Sciences, Hefei Institute of Stem Cell and Regenerative Medicine, Guangzhou Institutes of Biomedicine and Health, Chinese Academy of Sciences, Guangzhou Medical University, 510530 Guangzhou, China. [2] Guangzhou Regenerative Medicine and Health Guangdong Laboratory, Guangdong Provincial Key Laboratory of Stem Cell and Regenerative Medicine, Guangdong Provincial Key Laboratory of Biocomputing, Guangzhou Institutes of Biomedicine and Health, Chinese Academy of Sciences, 510530 Guangzhou, China. [3] Institute of Stem Cell and Regeneration, Chinese Academy of Sciences, 100101 Beijing, China. [4] University of Chinese Academy of Sciences, 100049 Beijing, China. [5] Department of Biology, Southern University of Science and Technology, 518055 Shenzhen, China. [6] Key Laboratory of Genomic and Precision Medicine, Beijing Institute of Genomics, Chinese Academy of Sciences, 100101 Beijing, China. These authors contributed equally: Jiao Li, Kaimeng Huang, Gongcheng Hu. Correspondence and requests for materials should be addressed to H.Y. (email: yao_hongjie@gibh.ac.cn)

During the last several decades, great strides have been made in understanding and deciphering the sophisticated higher-order chromatin architecture of mammalian cells[1–3]. Recent progress has shown that the mammalian genome is organized into structural topologically associated domains[4], which the insulator protein CTCF partitions at the boundaries of such domains[5–7] mainly through its zinc finger (ZF) DNA binding domain[8,9]. The precise nucleotide sequences at those loci are quite critical, as inversion, deletion or mutation of CTCF binding have been reported to affect higher-order chromatin organization and transcriptional regulation[6,10–14], diseases[15] or tumorigenesis[5,16]. Cell-to-cell variation of gene expression seems to be controlled by CTCF-mediated promoter—enhancer interactions, suggesting that the dynamics of CTCF-mediated higher-order chromatin structure is important, although the mechanisms are poorly understood[17].

Alternative splicing is the process by which splice sites in primary transcripts are differentially selected to produce structurally and functionally distinct mRNA and protein isoforms[18]. It provides a powerful mechanism to expand the functional and regulatory capacity of metazoan genomes. Genome-wide studies estimated that 90–95% of human genes undergo alternative splicing[19,20], and a subset of alternative splicing events has been identified to regulate development[21], tissue identity[22], pluripotency[23], and tumor proliferation[24]. Yet, the role of alternative splicing in chromatin organization has not been widely explored, and it may be an important factor, as it may control chromatin architecture to modulate regulatory pathways that can affect cell fate or function.

Previous studies have focused on investigating the roles of the canonical isoform of CTCF in gene regulation and genome organization[13,25]; nothing has been reported about the alternative splicing of CTCF and the roles of spliced isoforms in regulating higher-order chromatin structure and cellular function. In this study, we confirm a short CTCF (CTCF-s) isoform in the human genome. CTCF-s has the ability to compete with CTCF binding. Importantly, our data indicate that at those loci where CTCF and CTCF-s compete there is decreased level of cohesin, and alteration in CTCF-mediated chromatin looping. CTCF-s gain-of-function leads to the activation of *IFI6*, which ultimately results in increased cell apoptosis. CTCF-s activates *IFI6* expression by disrupting CTCF chromatin insulator function within the *IFI6* gene, facilitating an abnormal long-range interaction between an *IFI6* distal enhancer and its promoter. Together, these findings demonstrate how alternatively spliced versions of important architectural proteins can have key effects on cell apoptosis by altering genome architecture.

## Results

### Identification of alternatively spliced CTCF-s isoform in humans.
From our own RNA-seq data in several human cell lines (HeLa-S3 and 293T cells), we found that CTCF might have an alternatively spliced short isoform in which two exons (exons 3 and 4) are skipped, producing a truncated CTCF protein with an alternative translation start site at exon 5 [26]. This putative shorter isoform (we termed CTCF-s) lacks the sequence encoding the N-terminal domain plus 2.5 zinc fingers (ZFs), but still effectively contains eight intact ZFs and a full length C-terminal domain (Fig. 1a). To verify the presence of this short isoform, we performed nested PCR and obtained two fragments across exons 2 and 5, suggesting exon skipping occurs within this region (Fig. 1b, c). However, we only observed one fragment when the primers were between exons 3 and 5, which indicated that exons 3 and 4 only presented one isoform of CTCF (Fig. 1c). Sanger

sequencing confirmed the lower band as CTCF-s, which had no exons 3 and 4 (Fig. 1c, lane 4, Fig. 1d).

To demonstrate the ubiquity or not of CTCF-s, we measured the levels of CTCF-s mRNA (by TaqMan qPCR) and protein in different human cell lines and found that CTCF-s was ubiquitously expressed in all tested human cell lines, but with cell-type specific variations in the ratios of long and short isoforms (Fig. 1e, f and Supplementary Figure 1). To further demonstrate the existence of the short isoform at the protein level, we designed shRNAs specifically targeting only one of the two isoforms (shCTCF-s#1 for CTCF-s, shCTCF#1 and shCTCF#2 for CTCF) or both isoforms (shCTCF-both#1 and shCTCF-both#2 for both CTCF and CTCF-s) (Supplementary Figure 1a). The knockdown efficiency was assayed by TaqMan qPCR, which confirmed effective reductions in the levels of the targeted mRNA (Fig. 1g). Consistent with the mRNA results, Western blot assay revealed that isoform-specific shRNAs significantly decreased the expression of the corresponding isoform without altering the expression of the other isoform (Fig. 1h). Collectively, these data indicate the existence of an alternatively spliced short isoform (CTCF-s) in humans.

### CTCF-s shares similar 2D-chromatin functions with CTCF.
Because of the importance of canonical CTCF in regulating gene expression and chromatin organization[27] and the similarity between CTCF-s and CTCF in ZF composition, we were curious to know whether CTCF-s also played a role in chromatin organization. To test this, we determined their genome-wide DNA occupancy by using chromatin immunoprecipitation followed by high-throughput sequencing (ChIP-seq). Due to the lack of a specific antibody against CTCF-s, we introduced a FLAG-tagged CTCF and CTCF-s into HeLa-S3 cells, respectively (Supplementary Figure 2a-c). Subcellular localization analysis showed that FLAG-CTCF and FLAG-CTCF-s were both mainly located in the chromatin-bound fraction (Supplementary Figure 2d). FLAG ChIP-seq for CTCF identified 20,010 peaks, and 80.7% of these peaks overlapped with CTCF ChIP-seq from ENCODE (Supplementary Figure 2e), suggesting good data quality for FLAG ChIP-seq, although the overall number of peaks was lower than the 79,639 peaks observed when using an antibody directly targeting CTCF. To increase peak numbers, we therefore tried the biotin-tag system to further investigate the protein—DNA interaction mediated by CTCF or CTCF-s. This system has been reported to be an effective method for studying protein—protein and protein—DNA interactions in mouse embryonic stem cells[28]. We confirmed the in vivo biotinylation of CTCF and CTCF-s by Western blot (Supplementary Figure 3). Subsequently, we performed biotin ChIP-seq experiments for both CTCF and CTCF-s, and we identified 72,937 binding sites for CTCF and 41,362 binding sites for CTCF-s, respectively (Fig. 2a). Moreover, 86.86% (51,077/58,806) of CTCF peaks from ENCODE overlapped with our biotin-enriched CTCF peaks (Fig. 2a), suggesting high quality for our biotin ChIP-seq data. We next compared CTCF-s peaks with CTCF peaks generated from our biotin ChIP-seq. Our data showed that 68.8% of CTCF-s peaks overlapped with CTCF peaks (Fig. 2a, b), implying CTCF-s might have overlapping functions with CTCF in the genome.

Recent crystal structure analysis of CTCF ZF1-11 has shown that the core 15 bp DNA binding motif was mainly specified by ZF3-7 [8,9]. As CTCF-s has lost ZF1-3, we were interested to know if the DNA consensus motif was also altered. De novo motif discovery indeed reported a 2 bp truncation in the consensus motif, which are the base pairs that ZF3 makes contact with[8,9] (Fig. 2c). This has two potential consequences; the first consequence is that CTCF-s may have access to CTCF-s-specific

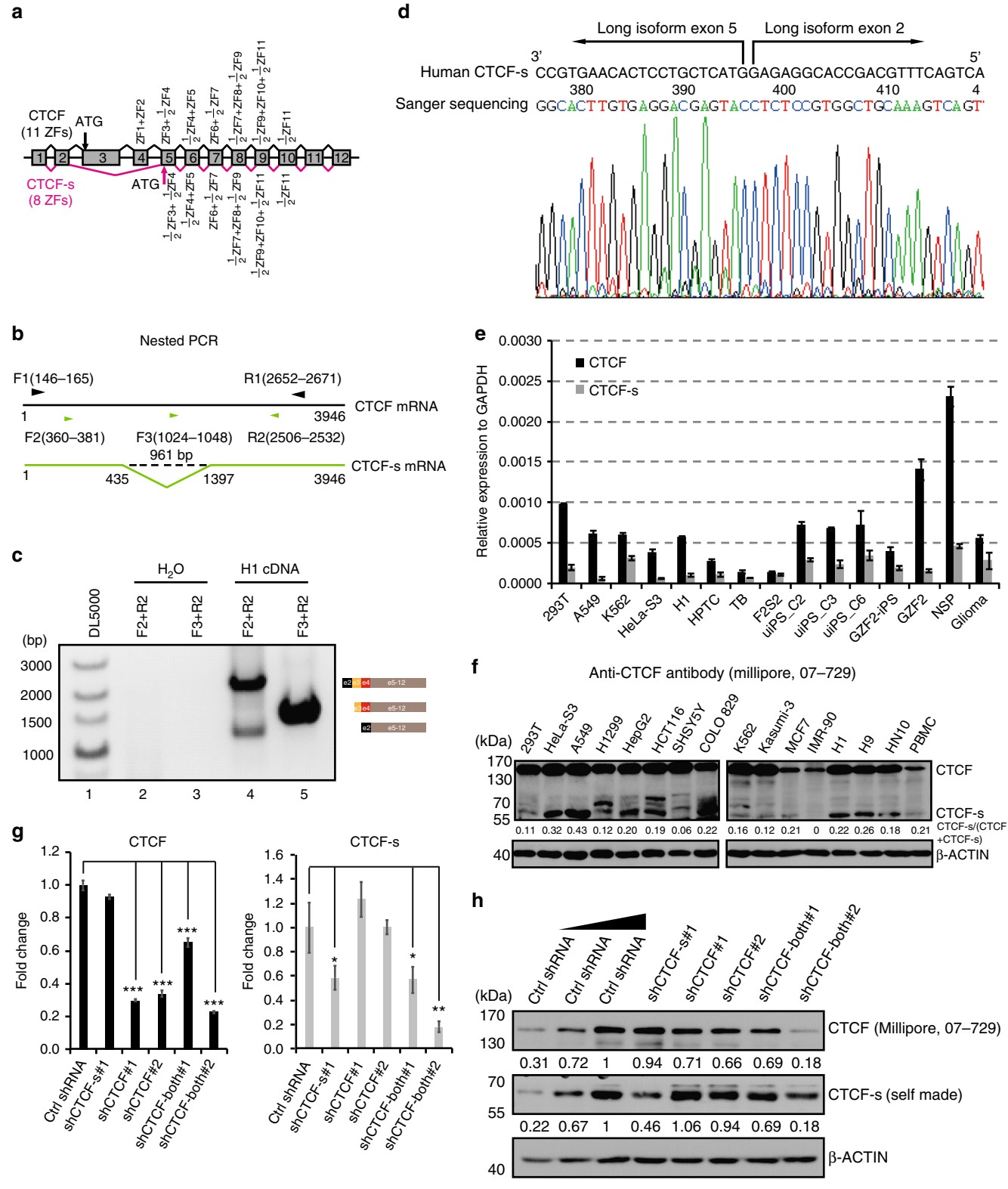

binding motifs that would normally be unfavorable for CTCF binding, and the second consequence is the possibility of direct DNA competition between CTCF and CTCF-s as the CTCF-s motif is a subset of the CTCF motif. Consistently, gel mobility shift assay showed that CTCF or CTCF-s could quantitatively compete for DNA binding when the two proteins were incubated with the same DNA probe (Fig. 2d, e).

**CTCF-s competes with canonical CTCF binding in the genome.** Based on the above in vitro and in vivo data, we proposed that CTCF-s could compete with CTCF in living cells. To explore this possibility, we performed ChIP-seq experiments for CTCF using an N-terminal-specific antibody (Supplementary Data 3), which specifically detected the long isoform of CTCF and allowed us to probe the changes in CTCF binding when CTCF-s was

**Fig. 1** Identification of an alternatively spliced CTCF-s isoform in the human genome. **a** Schematic representation of exons of the human CTCF gene. Transcripts including alternative exons 3 and 4 encoded the widely expressed canonical form of CTCF (top), and the CTCF-s isoform excluded exons 3 and 4. The canonical form of CTCF contained 11 ZFs, while CTCF-s effectively lacked the N terminal and 3 ZFs. **b** Strategy for detecting the CTCF short isoform. Black line represents the full length of CTCF mRNA (1−3946), and green line represents CTCF-s mRNA (1−435 joined with 1397−3946). Primer information was indicated based on the canonical long isoform. Convergent black arrows showed the positions of the first nested primers; green arrows indicated the position of the second nested PCR primers, with F2 and R2 for amplifying both CTCF and CTCF-s, F3 and R2 for amplifying CTCF only. **c** Nested PCR was used to validate the existence of the short isoform with the primers from (**b**). The constitution of each PCR product is presented on the right. **d** Chromatogram from Sanger sequencing of the lower band of panel (**c**), showing the junction at exons 2 and 5. **e** TaqMan RT-qPCR analysis of the relative expression levels of CTCF and CTCF-s in various human cell lines. Data were shown as mean ± s.d., $n = 3$ technological repeats. **f** Western blot of CTCF and CTCF-s in different human cell lines with anti-CTCF antibody (Millipore, 07-729). The locations of the full-length CTCF and CTCF-s were indicated. **g** RT-qPCR analysis of two different CTCF isoforms after specific shRNA knockdown. Data were shown as mean ± s.d. from $n = 3$ independent technological repeats with the indicated significance by using a two-tailed Student's $t$ test, $*p < 0.05$, $**p < 0.01$, $***p < 0.001$. **h** Western blot analysis of two different CTCF isoforms after specific shRNA knockdown. Source data are provided as a Source Data file

overexpressed, and compared CTCF occupancy in FLAG control and FLAG-CTCF-s-overexpressing cells. We firstly compared our CTCF ChIP-seq data with three published CTCF ChIP-seq data from ENCODE and found that at least 90.3% of published CTCF binding peaks overlapped with our CTCF peaks (Supplementary Figure 4).

We then compared the CTCF ChIP-seq signals between FLAG control and FLAG-CTCF-s-overexpressing cells. The majority of CTCF binding was not affected by the CTCF-s overexpression (cluster 2 in Fig. 2f, g); however, we interestingly found that CTCF-s overexpression resulted in significant reductions in binding at 24.2% (15,095/62,410) of overlapped CTCF sites between FLAG control and FLAG-CTCF-s-overexpressing samples (cluster 1 in Fig. 2f, g). Genomic tracks and ChIP-qPCR results confirmed that CTCF-s reduced CTCF occupancy, for example, at the *MDM2/CPM* loci (Fig. 2i, j), suggesting that CTCF-s could affect the DNA binding pattern of CTCF. We next asked why CTCF peaks in cluster 1 but not in cluster 2 could be competed by CTCF-s. We compared the peaks in clusters 1 and 2 with the CTCF-s binding peaks from the CTCF-s-specific biotin ChIP-seq; we found that peaks in cluster 1 were bound by CTCF-s, while CTCF-s binding in cluster 2 was much weaker (Fig. 2f, h). This suggests that CTCF and CTCF-s are competing at specific sets of genomic binding sites.

We next wondered whether the association of CTCF-s or CTCF with chromatin is different. To address this question, fluorescence recovery after photobleaching (FRAP) was carried out to determine the protein:DNA dynamics in living cells. We expressed CTCF-s-GFP or CTCF-GFP fusion proteins in 293T cells and then compared the FRAP curves obtained with the two isoforms. Consistent with the in vitro EMSA results (Fig. 2d, e), CTCF-s displayed faster FRAP recovery than CTCF (Fig. 2k), which indirectly indicated that there was a weaker CTCF-s:DNA dynamics than CTCF:DNA.

**CTCF-s competition decreases cohesin binding**. Since CTCF affects 3D chromosome architecture[6], we wondered whether CTCF-s competition could alter CTCF-mediated chromatin looping. Considering that the cohesin complex associates with CTCF to form higher-order chromatin organization[27], we first determined if there was a change in cohesin occupancy upon CTCF-s gain-of-function. We performed RAD21 ChIP-seq and identified 80,835 RAD21 binding peaks in FLAG control cells, closely matching an ENCODE dataset (Supplementary Figure 5a). Consistent with previous studies, 58% (46,845/80,835) of RAD21 binding sites overlapped with 75% (46,845/62,410) of CTCF binding sites (Fig. 3a)[27]. Surprisingly, in contrast to the overall ratio of co-binding sites between cohesin and CTCF, for those CTCF peaks that were reduced upon CTCF-s overexpression, the vast majority of these sites (14,646/15,278, 95.9%) were also

co-bound by RAD21 in control cells (Fig. 3b). We thus wondered if the decrease in CTCF binding intensity was caused by CTCF-s competing with CTCF/cohesin binding. To address this question, we compared RAD21 binding intensity between FLAG control and FLAG-CTCF-s-overexpressing cells. We found that the number of significantly decreased RAD21 peaks was considerably larger than the number of significantly increased peaks (6434 down vs. 66 up sites) (Supplementary Figure 5b), indicating that CTCF-s disrupted cohesin binding in the genome. Moreover, the majority (5414/6434) of these significantly decreased RAD21 peaks also overlapped with CTCF peaks (Supplementary Figure 5c). We therefore analyzed the relationship between the 14,646 significantly decreased CTCF peaks (CTCF-s/CTCF competition sites) upon CTCF-s gain-of-function and 5414 significantly decreased RAD21 peaks. We classified these binding sites into three clusters, in which cluster 1 (1004) and cluster 2 (4410) represented significantly decreased RAD21 binding sites; cluster 2 and cluster 3 (10,236) represented significantly reduced CTCF binding sites. We found both CTCF and RAD21 were decreased in these three clusters upon CTCF-s overexpression (Fig. 3c). We next asked why decreased CTCF binding led to decreased cohesin binding; we therefore investigated the physical interaction between cohesin and CTCF/CTCF-s. Consistent with previous studies, CTCF physically interacted with cohesin. However, to our surprise, CTCF-s displayed a weaker interaction with the cohesin subunit RAD21 than CTCF (Supplementary Figure 5d). Together, we conclude that at those sites where CTCF-s was competing with CTCF, cohesin binding was similarly reduced.

**CTCF-s competition alters CTCF-mediated chromatin contacts**. The reduction in CTCF binding and cohesin binding in response to CTCF-s competition implies that these loci may also display decreased looping intensity[29,30]. To determine CTCF-mediated chromatin looping and the dynamics of CTCF-mediated chromatin looping upon CTCF-s competition, we performed CTCF HiChIP experiments[31] in FLAG control and in FLAG-CTCF-s-overexpressing cells. The quality of CTCF HiChIP libraries generated from each experiment was good, as nearly 40% of the reads represented unique paired-end tags (Supplementary Data 4) and biological replicates showed good correlation (Pearson $R = 0.854$ for FLAG control replicates and $R = 0.958$ for FLAG-CTCF-s overexpression replicates, respectively) (Supplementary Figure 6a, b).

We then examined the dynamics of chromatin looping mediated by CTCF upon CTCF-s gain-of-function. We firstly inspected the interaction matrix at progressively higher resolutions, and found that chromatin features were similar to Hi-C interaction map at 500, 25, and 10-kb resolution (Fig. 3d). We next compared the chromatin loops in FLAG control and

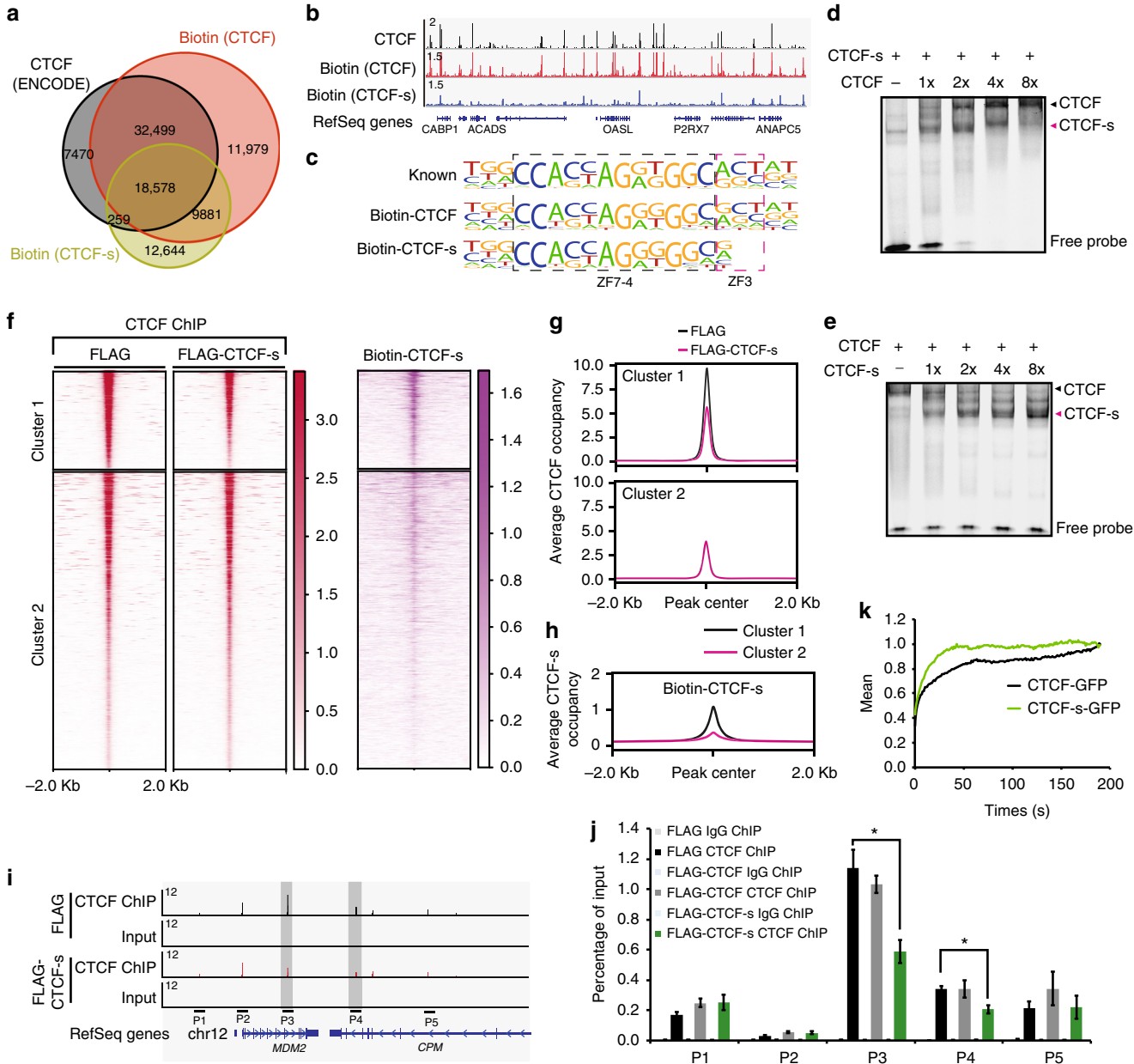

**Fig. 2** CTCF-s has its specific genomic binding but also competes with CTCF in binding to their common targets. **a** Venn diagram showing the overlap of biotin-enriched CTCF or CTCF-s peaks with a normal CTCF ChIP-seq from ENCODE (GSE33213). **b** Genomic views of ChIP-seq intensities of CTCF (ENCODE), biotin ChIP-seq for biotin-CTCF and biotin-CTCF-s in HeLa-S3 cells at chromosome 12:121,055,606−121,842,881. **c** Consensus DNA binding motif generated from de novo motif discovery. The motif inside the black dotted box is bound by ZF4-7 of CTCF and the motif inside the magenta dotted box is bound by ZF3 of CTCF. **d** Electrophoresis mobility shift analysis (EMSA) showing DNA binding ability of CTCF and CTCF-s as the concentrations of CTCF increase. **e** EMSA showing DNA binding ability of CTCF and CTCF-s with increasing amounts of CTCF-s. **f** Heatmap showing the comparison of CTCF binding occupancies between FLAG control and FLAG-CTCF-s overexpression. Cluster 1 shows the significantly reduced occupancy of CTCF after CTCF-s overexpression; Cluster 2 shows no difference in the levels of CTCF genomic occupancy with or without CTCF-s overexpression. Heatmap on the right showing the biotin ChIP-seq signal for CTCF-s. **g** Tag density pileup for all CTCF peaks from clusters 1 and 2, respectively. **h** Tag density pileup for biotin-CTCF-s peaks from clusters 1 and 2, respectively. **i** Genomic view showing ChIP primers P1−P5 at *MDM2-CPM* loci in chromosome 12. Peaks marked by gray box represent two decreased CTCF peaks upon CTCF-s gain-of-function. **j** Verification of CTCF ChIP-seq data in chromosome 12 using ChIP-qPCR. IgG was used as a negative control. The locations of the primer pairs P1−P5 are indicated in panel (**i**). The data were represented as mean ± s.d. ($n = 3$ technological repeats) with the indicated significance by using a two-tailed Student's *t* test (*$p < 0.05$). The entire experiment was performed three times. **k** FRAP analysis of GFP-CTCF and GFP-CTCF-s after irreversible photobleaching. Source data are provided as a Source Data file

FLAG-CTCF-s-overexpressing cells (see Methods, Supplementary Data 4). We found that 5986 loops were significantly decreased and 1080 loops were increased upon CTCF-s gain-of-function (Fig. 3e). In addition, decreased loops were present at stronger interacting intensity than those loops that were not affected by CTCF-s overexpression, while increased loops were present at the weakest interacting intensity among the three categories before CTCF-s gain-of-function (Fig. 3f). We next set out to see how CTCF-s competition affected CTCF-mediated chromatin looping. We divided the chromatin loops into three categories: CTCF

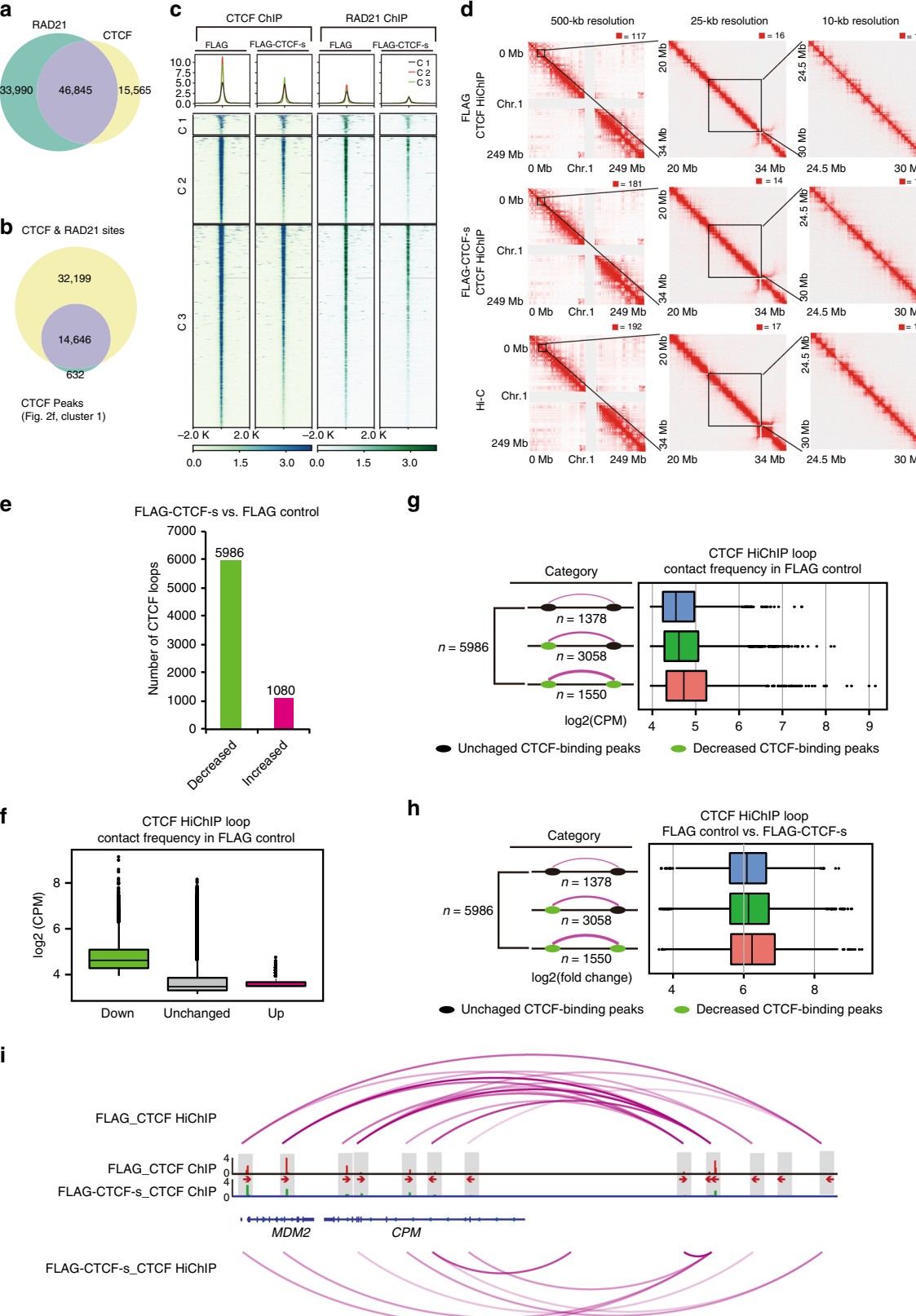

binding with no change, CTCF binding reduction at one of the anchors, and reduction at both anchors (Fig. 3g). In total, there were 5986 loops that decreased, of which 3058 loops decreased at only one anchor, 1550 decreased at both anchors, and 1378 loops where neither anchor was affected (Fig. 3g). Moreover, upon CTCF-s gain-of-function, loops with no CTCF binding alteration

displayed the weakest interaction frequency, loops with CTCF binding reduction at one anchor displayed the median interaction frequency, while loops with a decrease at both anchors showed the highest interaction frequency (Fig. 3g). We also analyzed the dynamics of the classes of loops when CTCF-s was overexpressed. We found that loop intensity decreased more in those cases where

**Fig. 3** CTCF-s competition causes the reduction of both CTCF and cohesin binding and alters CTCF-mediated chromatin looping. **a** Venn diagrams showing the overlap of RAD21 and CTCF peaks. **b** Venn diagrams showing the overlap of RAD21 binding sites with significantly decreased CTCF binding sites upon CTCF-s gain-of-function, from cluster 1 in Fig. 2f. **c** Heatmap and pileup for CTCF and RAD21 ChIP intensity in CTCF bound sites that decreased when FLAG-CTCF-s was overexpressed. **d** Raw interaction maps of CTCF HiChIP at a locus on chromosome 1 produced from FLAG control cells and cells with FLAG-CTCF-s overexpression and Hi-C data in HeLa-S3 cells drawn with the indicated resolutions and views. Numbers above the interaction maps correspond to maximum signal in the matrix. **e** Bar chart showing loop difference from HiChIP data in both FLAG control cells and cells with FLAG-CTCF-s overexpression. **f** Boxplot showing the distribution of loop intensities in average log2 counts per million (log2CPM) among the three loop classes (up, down and unchanged) from (**e**). The midline represents the second quartile, the bounds of box represent third quartile and first quartile. The whiskers indicate 1.5 × interquartile range and dots represent outliers. **g** Features of downregulated loops with distinct anchor features in FLAG control from (**f**). The loop anchor features were classified from the change of CTCF binding strength in FLAG control and FLAG-CTCF-s overexpressed samples. The midline represents the second quartile, while the bounds of box represent third quartile and first quartile, respectively. The whiskers indicate 1.5 × interquartile range and dots represent outliers. **h** Fold-change analysis of downregulated loops that were associated with distinct anchor features that were classified from FLAG control and FLAG-CTCF-s overexpressed samples after FLAG-CTCF-s gain-of-function. The midline represents the second quartile, the bounds of box represent third quartile and first quartile. The whiskers indicate 1.5 × interquartile range and dots represent outliers. **i** Example genomic view showing the changes of CTCF ChIP intensities and CTCF-mediated chromatin loops at the *MDM2/CPM* gene locus. The directionality of the motifs at CTCF loop anchors was indicated with red arrows

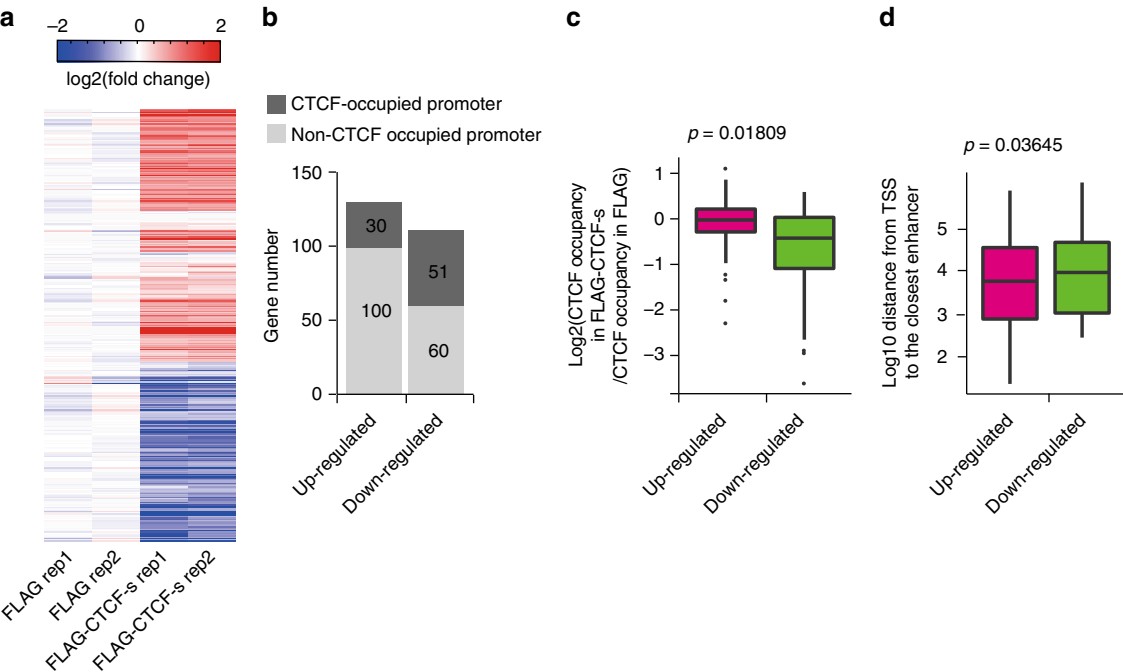

**Fig. 4** CTCF-s/CTCF competition and transcriptional regulation. **a** Hierarchical clustering of differentially expressed (DE) gene profiles after overexpression of FLAG-CTCF-s in HeLa-S3 cells. Fold-change was relative to the mean of the FLAG control. **b** Analysis of CTCF binding at promoters of DE genes, from panel (**a**). **c** CTCF-s downregulated genes tended to have decreased CTCF binding at their promoter regions. The center line represents the second quartile, the bounds of box represent third quartile and first quartile. The whiskers indicate 1.5 × interquartile range and dots represent outliers. **d** Upregulated genes by CTCF-s tended to have enhancers that were closer to the TSS than downregulated genes. The center line represents the second quartile, the bounds of box represent third quartile and first quartile. The whiskers indicate 1.5 × interquartile range and dots represent outliers. The statistical significance for the boxplots in (**c**, **d**) was assessed by Wilcoxon rank sum test

CTCF binding to both anchors was reduced, while the decrease was modest than that of CTCF binding reduction at either one anchor by CTCF-s gain-of-function; however, in those loops where CTCF binding was unaltered, loop intensity decreased the least (Fig. 3h). For example, decreased CTCF loop frequency at the *MDM2/CPM* locus was observed where CTCF bindings were reduced after CTCF-s overexpression (Fig. 3i, Supplementary Figure 6c). Overall, these observations suggest that the binding competition between CTCF-s on CTCF leads to decreased binding of cohesin complex, and subsequent alterations of CTCF-mediated chromatin looping.

**CTCF-s competition leads to downstream gene deregulation.** It has been reported that CTCF-mediated genome folding is related

to transcriptional regulation[32]. As CTCF-s competition can alter CTCF binding and CTCF-mediated chromatin looping, we hypothesized that this competition might alter gene expression. We therefore compared the transcriptional profiles upon ectopic expression of FLAG alone and FLAG-CTCF-s. Upon CTCF-s gain-of-function, 130 genes were upregulated and 111 genes were downregulated (Fig. 4a). Integration with CTCF ChIP-seq data revealed that nearly 46% (51/111) of the downregulated genes had CTCF bound within the promoter, as opposed to 23.8% (30/130) of the upregulated genes (Fig. 4b). Moreover, consistent with a recent report about the link between auxin-induced degradation of CTCF and transcriptional regulation[29], of those downregulated genes, we also observed significantly decreased CTCF binding at promoter regions (Fig. 4c). While of those upregulated genes, we

noticed that the distance between promoter and flanking enhancer was much closer (Fig. 4d).

**CTCF-s promotes cell apoptosis through *IFI6*.** Since our evidence indicates that CTCF-s modulates CTCF-mediated

chromatin organization and transcriptional regulation, we were curious to know whether CTCF-s participated in any cellular processes. Gene ontology (GO) analysis indicated that CTCF-s upregulated genes were involved in type I interferon signaling and apoptosis (Fig. 5a), and RT-qPCR experiments showed that the

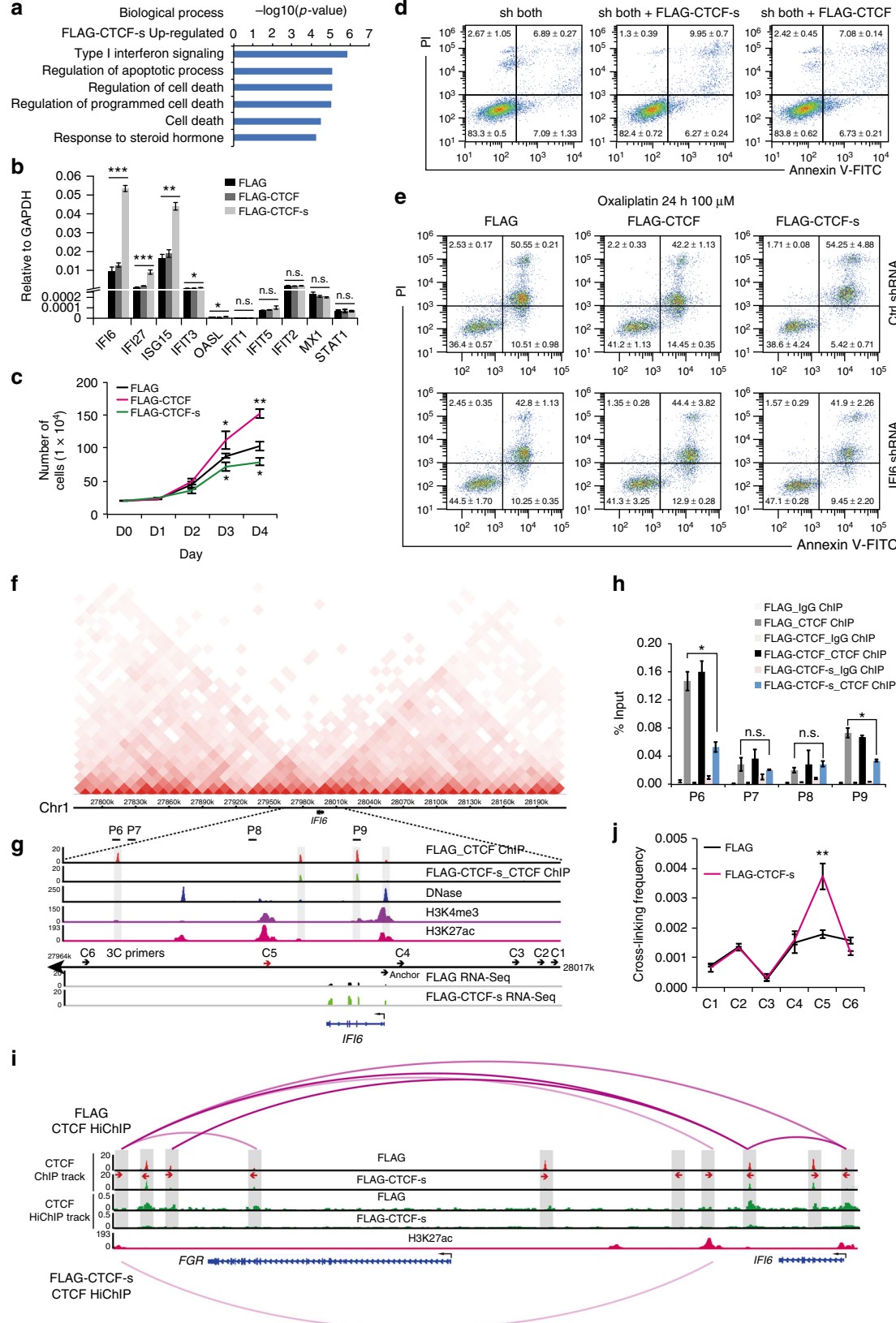

expression of type I interferon genes, *IFI6*, *IFI27* and *ISG15* were only stimulated by CTCF-s, but not by CTCF (Fig. 5b), which was further confirmed by RNA-seq from specific knockdown of CTCF and/or CTCF-s (Supplementary Figure 7). We examined cell viability and found that CTCF overexpression promoted cell proliferation[33], while CTCF-s inhibited cell proliferation (Fig. 5c). Consistent with the above data, FACS results showed that CTCF-s overexpression promoted cell apoptosis (Supplementary Figure 8 and 9a). To further confirm the roles of CTCF/CTCF-s on apoptosis, we specifically knocked down these two isoforms simultaneously, and then overexpressed synonymously mutated CTCF and CTCF-s, respectively, which can overcome the effect of the shRNA-mediated knockdown (Supplementary Figure 10). Our results further showed that the rescue of CTCF-s can promote cell apoptosis ($9.95 \pm 0.7$ vs. $6.89 \pm 0.27$, $p = 0.0021$, Student's $t$ test) while rescue of CTCF did not affect apoptosis ($7.08 \pm 0.14$ vs. $6.89 \pm 0.27$, $p = 0.34$, Student's $t$ test) (Fig. 5d). Moreover, we introduced oxaliplatin, a DNA damage reagent[34,35], to induce apoptosis and accelerate the apoptotic process and then investigated the effect of CTCF-s. Consistently, after apoptosis acceleration, we observed that CTCF-s promoted apoptosis mainly at the late stage (Supplementary Figure 9b, c). Collectively, our data suggested that CTCF-s, but not CTCF, promoted cell apoptosis. As type I interferon-signaling pathway can induce apoptosis[36], we knocked-down *IFI6*, an important downstream target gene induced by the type I interferon-signaling pathway, in FLAG-CTCF or FLAG-CTCF-s-overexpressing cells. Our results showed that apoptosis induced by CTCF-s was inhibited by *IFI6* silencing even after DNA damage ($54.25 \pm 4.88$ vs. $41.9 \pm 2.26$, $p = 0.083$, Student's $t$ test) (Fig. 5e), suggesting that *IFI6*-type I interferon-signaling pathway could be an important pathway for CTCF-s-mediated apoptosis.

**CTCF-s activates *IFI6* by altering enhancer−promoter contact.** To further investigate the role of CTCF-s in activating *IFI6* expression, we considered whether the stimulation of *IFI6* was related to the alteration of CTCF-mediated chromatin looping at this locus. From Hi-C interaction map data, we noticed that the *IFI6* gene was located near the boundary of two flanking sub-TADs (Fig. 5f). In support, our CTCF ChIP-seq data showed that the *IFI6* gene was surrounded by multiple CTCF binding sites, and CTCF ChIP-seq data indicated that CTCF-s gain-of-function disrupted the occupancies of CTCF around the *IFI6* gene (Fig. 5g), which was further validated by ChIP-qPCR (Fig. 5h). We then investigated the dynamics of CTCF-mediated long-range interactions around this locus. As described above, we observed that the global interaction patterns were disrupted after CTCF-s

gain-of-function (Fig. 5i, Supplementary Figure 6d). We wondered how this global disruption of CTCF binding at the *IFI6* locus affected *IFI6* expression. The current understanding of chromatin structure led us to hypothesize that *IFI6*, being located near the boundary, might be activated via the loss of a neighborhood boundary with aberrant activation by enhancers located outside of the chromatin neighborhood. To confirm this, we analyzed the promoter marker (H3K4me3) and enhancer markers (H3K27ac and DNase) at the *IFI6* loci. As shown in Fig. 5g, we observed several likely enhancers (one proximal upstream and two distal downstream enhancers) at this locus. Moreover, the enhancers are located in different sub-TADs to *IFI6*. To answer whether the disruption of CTCF loops caused abnormal enhancer−promoter contact to activate *IFI6* transcription, we performed chromosome conformation capture (3C) to query the relative frequencies with which the *IFI6* promoter interacted with the flanking enhancers or with nearby control sites. Upon CTCF-s gain-of-function, 3C revealed no change between the *IFI6* promoter and its proximal enhancer, which are just ~1 kb apart (Fig. 5g). On the contrary, we observed a more frequent interaction between the *IFI6* promoter and its distal intragenic enhancer (~13 kb apart) after CTCF-s gain-of-function (Fig. 5g, j), suggesting CTCF-s gain-of-function could activate *IFI6* gene expression by changing chromatin conformation and higher-order chromatin structure at this locus.

**Discussion**
The mammalian genome is organized into different degrees of chromatin architecture. The development of ChIA-PET and recently developed HiChIP technology has greatly improved the resolution of protein-centric higher-order chromatin organization[5,31]. Although the TADs in cells are quite stable, the dynamics of chromatin looping mediated by CTCF and other factors that form the long-range chromatin interactions vary between different cells even within one cell population[17].

CTCF is a key factor in mediating long-range chromatin interactions, yet most studies on CTCF focus on the canonical long isoform of CTCF and its many known biological functions[4,5,14,37–40]. Two recent investigations about the crystal structure of CTCF ZFs specify the nucleotides bound by each ZFs[8,9]. Consistently, biotin-CTCF specifically binds the 15 bp core DNA motif, and biotin-CTCF-s, which lacks 7 aa (HKCPDCD) of the 24-aa ZF3, no longer recognizes the two-core motif (C/G T/C) for ZF3 and only preserves G/A with decreased specificity. Our data suggest that CTCF-s competes with CTCF to reduce cohesin binding, and further disrupts chromatin looping mediated by CTCF, providing an additional eviction mechanism

---

**Fig. 5** CTCF-s activates *IFI6* by disrupting CTCF-mediated looping and establishing distal enhancer−promoter contacts. **a** GO analysis of upregulated genes after CTCF-s gain-of-function. **b** Relative expression of interferon-signaling-associated genes after CTCF and CTCF-s gain-of-function, respectively (mean ± s.d., two-tailed Student's $t$ test; $n = 3$ biological replicates, each biological replicate had three technical replicates). **c** Growth curves of HeLa-S3 cells with gain-of-function of FLAG, FLAG-CTCF and FLAG-CTCF-s, respectively. Data were shown as mean ± s.d. from three independent experiments with the indicated significance by using a two-tailed Student's $t$ test. **d** Effects of CTCF and CTCF-s overexpression on apoptosis in HeLa-S3 cells after knocking-down endogenous CTCF and CTCF-s. Data were shown as mean ± s.d., $n = 3$ biological replicates. **e** Apoptosis analysis of HeLa-S3 cells overexpressing FLAG, FLAG-CTCF and FLAG-CTCF-s following *IFI6* knocking-down and treatment with oxaliplatin for 24 h. Data were shown as mean ± s.d. from two independent biological replicates. **f** Hi-C interaction map at *IFI6* locus. **g** Genomic view showing CTCF ChIP-seq and RNA-seq data from FLAG and FLAG-CTCF-s, H3K4me3, H3K27ac and DNase-seq data at the *IFI6* locus. **h** Verification of CTCF ChIP-seq data in *IFI6* locus by using ChIP-qPCR. IgG was used as a negative control. The primers (from P6 to P9) were indicated in (**g**). The data were represented as mean ± s.d. with the indicated significance from a two-tailed Student's $t$ test, $n = 3$ technological replicates. **i** CTCF HiChIP interactions at *IFI6* locus displayed with CTCF ChIP-seq and one-dimensional track of CTCF HiChIP in FLAG and FLAG-CTCF-s overexpressing cells. The directionality of the motifs at CTCF loop anchors was indicated with red arrows and gray bars. **j** 3C assay measuring the crosslinking frequency in FLAG and FLAG-CTCF-s overexpressing HeLa-S3 cells. *IFI6* promoter (anchor) and six complementary primers (C1 to C6 in (**g**)) were tested by 3C-qPCR. The data were represented as mean ± s.d. with the indicated significance by using a two-tailed Student's $t$ test, $n = 3$ technological replicates. For statistical significance, *$p < 0.05$, **$p < 0.01$, ***$p < 0.001$, n.s. not significant. Source data are provided as a Source Data file

for CTCF-mediated chromatin looping that is not triggered by mutations of CTCF binding sites or alteration of CTCF binding orientation[5,6]. It was suggested that recruitment of cohesin to most CTCF binding sites is in a CTCF-dependent manner[41,42]. Consequently, the reduction of cohesin could be explained if CTCF-s is impaired to recruit CTCF/cohesin, and so competes away CTCF and cohesin. Indeed, the interaction of CTCF and cohesin seems much stronger than that of CTCF-s and cohesin (Supplementary Figure 5d), which might be another reason to characterize the decreases of cohesin- and CTCF-binding in their common targets and to explain why CTCF-s gain-of-function changes CTCF-mediated chromatin looping in the genome.

Unlike the overexpression of CTCF that promotes the proliferation of HeLa-S3 cells[33], ectopically enhanced CTCF-s expression instead inhibited the proliferation of HeLa-S3 cells and resulted in apoptosis by activating *IFI6* expression, which was associated with the type I interferon-signaling pathway[43]. Mechanistically, CTCF-s competes with CTCF to disrupt the chromatin architecture, potentiating chromatin remodeling and aberrant promoter−enhancer interactions, and thus can modulate gene transcription.

In this study we have mainly focused on the roles of CTCF-s in the context of canonical CTCF binding; however, there were 12,644 CTCF-s binding sites that did not overlap with CTCF peaks (Fig. 2a). This intriguing observation suggests that in addition to its competitive function, CTCF-s binds to genomic regions independently of CTCF and provides an additional layer of genome organization. Further, the splicing pattern of *CTCF* varies in different cell types, providing a diversity of CTCF-mediated higher-order chromatin organization possibilities in different cell types. Therefore, mechanistic investigation of the transcriptional circuit of CTCF-s and characterization of the biological or pathological functions, especially in cancer, of this currently reported CTCF-s short isoform will need to be addressed in the future.

In summary, we conclude that ectopic expression of CTCF-s can disrupt the chromatin architecture maintained by CTCF. Therefore, CTCF-s may act as a modulator or "fine-tuner" of CTCF activity, balancing CTCF-mediated chromatin organization and transcriptional regulation. The competition between CTCF and CTCF-s ultimately leads to an effect on cell phenotype as the cells activate an apoptotic pathway. Our study demonstrates the importance of alternatively spliced isoforms in chromatin and cellular activity.

## Methods

**Plasmid construction**. CTCF and CTCF-s cDNAs were cloned into **pSin-FLAG** vector. Lentiviral shRNA constructs for CTCF and CTCF-s were obtained by cloning shRNA oligos into the **pLKO.1** vector. All the shRNA targeting sequences used in this study were described in Supplementary Data 1. All the constructs were confirmed by Sanger sequencing.

**Cell culture**. HeLa-S3 (ATCC, CatLog: ATCC® CCL-2.2), 293T (ATCC, CatLog: ATCC® CRL-11268™), Guangzhou Fibroblast #2 (GZF2), IMR-90 (ATCC, CatLog: ATCC® CCL-186), MCF7 (ATCC, CatLog: ATCC® HTB-22), H1299 (ATCC, CatLog: ATCC® CRL-5803), HepG2 (Cell Resource Center of Shanghai Institutes for Biological Sciences, Chinese Academy of Sciences), HCT116 (ATCC, CatLog: ATCC® CCL-247), SHSY5Y (Dr. Fei Lan's laboratory) cells were cultured in DMEM/High Glucose (Hyclone) supplemented with 10% FBS. Kasumi-3 cells (ATCC, CatLog: ATCC® CRL-2725) were cultured in DMEM/high glucose (Hyclone) supplemented with 20% FBS. K562 (ATCC, CatLog: ATCC® CCL-243), A549 (ATCC, CatLog: ATCC® CCL-185), COLO 829 (ATCC, CatLog: ATCC®CRL-1974) and peripheral blood mononuclear cells (self-separated primary cells) were cultured in 1640 medium (Hyclone) supplemented with 10% fetal bovine serum (FBS). Human proximal tubule cells (self-separated primary cells) were cultured in DMEM/F12 medium (Hyclone) supplemented with 10% FBS, 0.1 mM non-essential amino acid (NEAA), 1 mM L-GlutaMAX, 0.1 mM β-mercaptoethanol and REGM Renal Epithelial Cell Growth Medium SingleQuot Kit (Lonza, CatLog: CC-4127). Human ESC cell lines H1 (Wi Cell), H9 (Wi Cell), HN10 (Hainan Medical University), and uiPSCs (urine cells-derived induced

pluripotent cells) clones were cultured in mTeSR1 medium (STEMCELL Technologies) on matrigel (Corning)-coated plates. Neural stem progenitor cells were cultured in N2B27 medium of a 1:1 mixture of DMEM/F12 and Neurobasal medium supplemented with 1× NEAA, 1% N2 (Invitrogen), 2% B27 (Invitrogen), 5 µg ml−1 insulin, 2 µg ml−1 heparin, 100 µM β-mercaptoethanol, 5 µM SB431542 and 5 µM Dorsomorphin. Cultured cells were maintained at 37 °C in a 5% $CO_2$ incubator.

**Generation of in vivo biotinylated cell lines**. Firstly, lentivirus for lenti-birAV5, pSin-FLBio, pSin-FLBio-CTCF, and pSin-FLBio-CTCF-s was assembled with psPAX2, pMD2.G vectors. HeLa-S3 cells were then infected with lenti-birAV5 lentivirus and selected with 10 µg ml−1 of blastcidin for 5 days. The overexpression of birAV5 was detected by Western blot with anti-V5 antibody. Then the birAV5-expressing cells were infected with pSin-FLBio, pSin-FLBio-CTCF, and pSin-FLBio-CTCF-s lentivirus, respectively. Thereafter, the cells were selected with 4 µg ml−1 of puromycin for 5 days. Finally, the in vivo biotinylation of CTCF and CTCF-s were detected with Anti-BIOTIN antibody (Cell Signaling Technology, CatLog: #7075) with a dilution ratio of 1:1000.

**RNA extraction and qRT-PCR**. Total RNA was isolated with Trizol reagent (Invitrogen). cDNA was synthesized by using Oligo dT(18) primer (Takara) and reverse transcriptase (Toyobo). Real-time PCR was performed by using SYBR Green mix (Bio-Rad) in a CFX96 real-time PCR system (Bio-Rad) according to the manufacturer's instruction. Transcript levels were normalized to GAPDH level. CTCF isoforms were detected by using TaqMan probes. The primers used for qRT-PCR analysis are shown in Supplementary Data 2. Specifically, the two isoforms were analyzed via TaqMan qPCR with the following sequences. *GAPDH*: forward: 5′-GGT GGT CTC CTC TGA CTT CAA C-3′, reverse: 5′-GTT GCT GTA GCC AAA TTC GTT GT-3′, probe: FAM-ACC CAC TCC TCC ACC TTT GAC-MGB. *CTCF*: forward: 5′-CAG AGG TTA ATG CAG AGA AAG TG-3′, reverse: 5′-AAT GCC CTG CCA CAG AGA TG-3′, probe: FAM-TGA AGC CTC CAA AGC CAA CAA-MGB. *CTCF-s*: forward: 5′-TTT CCC TCC TCA AAC TGA CTT TG-3′, reverse: 5′-TGT CGC AGT CTG GGC ACT T-3′, probe: FAM-CCA CGG AGA GGT ACT C-MGB.

**Subcellular fractionation**. Whole-cell extracts of HeLa-S3 cells were separated into soluble supernatant and chromatin-containing fractions as described[44]. Briefly, harvested cells were suspended in 1.5 ml of ice-cold buffer A (10 mM 4-(2-hydroxyethyl)-1-piperazineethanesulfonic acid (HEPES) (pH 7.9), 1.5 mM $MgCl_2$, 10 mM KCl, 0.2 mM ethylenediaminetetraacetic acid (EDTA), 1 mM DL-Dithiothreitol (DTT), 1× protease inhibitor cocktail, 1 mM phenylmethanesulfonyl fluoride (PMSF)), incubated on ice for 15 min, and then vortexed for 10 s followed by adding NP-40 to a final concentration of 0.26%. The supernatant (cytosolic fraction) was collected by centrifuging at $1300 \times g$ for 30 min. The nuclear pellet was washed twice with buffer A, and resuspended in buffer C (20 mM HEPES, 25% glycerol, 0.3 M KCl, 1.5 mM $MgCl_2$, 0.2 mM EDTA, 1× protease inhibitor cocktails, 1 mM PMSF and 0.5 mM DTT), rotated at 4 °C for 15 min, and then centrifuged at $1000 \times g$ for 5 min. The supernatant was collected and designated as the nuclear fraction. To further separate the nuclear fraction into nucleoplasmic and chromatin-bound fraction, the nuclear fraction was washed twice in buffer A, and then lysed in buffer B (3 mM EDTA, 0.2 mM ethylene glycol-bis(2-aminoethyl-lether)-N,N,N′,N′-tetraacetic acid (EGTA), 1 mM DTT, 1× protease inhibitor cocktails, 1 mM PMSF). The supernatant was collected by centrifugation at $1700 \times g$ at 4 °C for 4 min and designated as nucleoplasmic fraction. Insoluble pellet was washed twice in buffer B, and the final insoluble pellet was resuspended in Laemmli buffer and sonicated for five cycles and was designated as chromatin-bound fraction.

**Western blot analysis**. The cells were resuspended in RIPA buffer (1% Triton X-100, 0.1% sodium dodecyl sulfate (SDS), 150 mM KCl, 50 mM Tris-HCl (pH 7.4), 1 mM EDTA, 1% sodium deoxycholate, 1 mM PMSF, and 1× protease inhibitor cocktails) and sonicated. Total soluble proteins were obtained by centrifugation at $15,294 \times g$ for 10 min. Samples were separated on SDS-PAGE gel and transferred onto a polyvinyl difluoride (PVDF) membrane (Millipore). The PVDF membrane was blocked with 5% milk in tris buffer saline (TBS)-T (TBS with 0.05% Tween-20). Immunoblot analysis was performed with the indicated antibodies. The antibodies and their corresponding dilution used in this experiment are listed in Supplementary Data 3.

**Immunofluorescence**. Cells growing on coverslips were washed three times with PBS, then fixed with 4% paraformaldehyde for 20 min at room temperature, and blocked with 0.1% Triton X-100 and 3% bovine serum albumin (BSA) for 30 min at room temperature. Then the cells were incubated with primary antibody for 2 h. After three washes with PBS-T, incubation with secondary antibodies for 1 h, cells were then incubated in 4′,6-diamidino-2-phenylindole (DAPI) for 5 min. Then the coverslips were mounted on the slides for observation on the confocal microscope (Zeiss LSM800). The antibodies used in this experiment are listed in Supplementary Data 3.

**Electrophoretic mobility shift assay (EMSA).** DNA oligos were synthesized by Guangzhou IGE Biotechnology. The forward probe sequence: 5′-(cy5)CCC, ATG, GCT, GGC, CAC, CAG, GGG, GCG, GCA, CAG, ACC-3′, the reverse probe sequence: 5′-GGT, CTG, TGC, CGC, CCC, CTG, GTG, GCC, AGC, CAT, GGG-3′. dsDNA probes were generated by mixing cy5-labeled forward and unlabeled reverse strands in 1× NEB buffer 2 and heating to 95 °C for 5 min and subsequent cooling to room temperature. EMSA reaction was carried out using 1× EMSA buffer (10 mM Tris-HCl (pH 8.0), 0.1 mg ml$^{-1}$ BSA, 50 μM ZnCl$_2$, 100 mM KCl, 10% (v/v) glycerol, 0.1% (v/v) IGEPAL CA-630 and 2 mM β-mercaptoethanol), 200 nM dsDNA and varying amount of proteins (purified from *Escherichia coli* BL21) for 25 min at room temperature in a dark room. After incubation, the protein−DNA complexes were loaded onto 6% native PAGE gels and run at 200 V for 30 min in 1× TG buffer (25 mM Tris-HCl (pH 8.0), 192 mM glycine) in a cold room. Bands were visualized using a Typhoon FLA-7000 PhosphorImager (FUJIFILM).

**Fluorescence recovery after photobleaching (FRAP).** For FRAP experiments, 293T cells were transiently transfected with GFP-tagged human CTCF (or CTCF-s) and grown overnight in cover glass chambers at a density of 2×10$^5$ in DMEM containing 10% FBS. FRAP experiments were carried out on a Zeiss 710 confocal microscope with a ×100/1.4 numerical aperture oil immersion objective, and the cells were kept at 37 °C using an air stream stage incubator. Bleaching was performed with a circular spot using the 488 and 514 nm lines from a laser operating at 100% laser power. A single iteration was used for the bleach pulse, and fluorescence recovery was monitored at low laser intensity (1.2%) at 1-s intervals. Data from at least three independent experiments were collected and used to generate corresponding average FRAP curves and normalized according to the manufacturer's instructions.

**Chromatin immunoprecipitation (ChIP).** ChIP was performed as previously described[45]. Briefly, crosslinked cells were sonicated and diluted tenfold with ChIP dilution buffer, and then incubated with protein A and protein G dynabeads (1:1 mix) and the indicated antibodies at 4 °C overnight. Antibody-bound DNA was subsequently washed with low salt wash buffer, high salt wash buffer, LiCl wash buffer once, respectively, and then TE wash buffer twice. ChIPed DNA was reverse-crosslinked and purified for DNA library construction followed by sequencing or ChIP-qPCR analysis. Primers used for ChIP-qPCR are listed in Supplementary Data 5.

For biotin ChIP-seq experiments, cells stably expressed biotin-alone or biotin-CTCF or biotin-CTCF-s were expanded and crosslinked with 1% formaldehyde. Crosslinked cells were sonicated and diluted tenfold with ChIP dilution buffer, and then incubated with M280 streptavidin dynabeads at 4 °C for overnight. Streptavidin dynabeads-bound DNA was subsequently washed twice with wash buffer 1 (2% SDS), once with wash buffer 2 (50 mM HEPES (pH 7.5), 1 mM EDTA, 500 mM NaCl, 0.1% sodium deoxycholate, 1% Triton X-100), once with wash buffer 3 (10 mM Tris-HCl (pH 8.0), 1 mM EDTA, 250 mM LiCl, 0.5% NP-40, 0.5% sodium deoxycholate) and then twice with TE wash buffer (10 mM Tris-HCl (pH 8.0), 1 mM EDTA). ChIPed DNA was reverse-crosslinked and purified for DNA library construction followed by sequencing.

**HiChIP experiments.** The HiChIP protocol was performed as previously described[31,46] with some modifications. In brief, up to 15 million crosslinked cells were resuspended in 500 μl of ice-cold Hi-C lysis buffer (10 mM Tris-HCl (pH 7.5), 10 mM NaCl, 0.2% NP-40, 1× Roche protease inhibitors) twice. The nuclei pellet was resuspended in 100 μl of 0.5% SDS and incubated at 62 °C for 10 min with no shaking or rotation and then the reaction was quenched with Triton X-100 at 37 °C for 15 min. *Mbo*I restriction enzyme (NEB, R0147) was added at 37 °C for 2 h to digest the nuclei, and heat inactivated at 62 °C for 20 min. After filling in the restriction fragment overhangs and marking the DNA ends with biotin, in situ contact was generated with proximity ligation. The nuclei with in situ generated contacts were pelleted at 2500 × *g* for 5 min at room temperature and nuclear pellet was resuspended in nuclear lysis buffer (50 mM Tris-HCl (pH 7.5), 10 mM EDTA, 1% SDS, 1× Roche protease inhibitor) for sonication and clarified by centrifugation at 16,100 × *g* at 4 °C for 15 min. Clarified samples were transferred to a new tube and diluted with ChIP dilution buffer (0.01% SDS, 1.1% Triton X-100, 1.2 mM EDTA, 16.7 mM Tris-HCl (pH 7.5), 167 mM NaCl) for ChIP procedures. ChIPed DNA was quantified by Qubit (Thermo Fisher) to estimate the amount of Tn5 (Illumina) needed to generate libraries at the correct size distribution. 150 ng of ChIPed DNA was taken into the biotin capture step and tagmented with Tn5. Finally, the tagmented DNA containing beads was PCR- amplified and size-selected with AMPure XP beads (Beckman). After size selection, libraries were quantified with qPCR against Illumina primers and/or bioanalyzer. Libraries were paired-end sequenced with read length of 150.

**Chromosome conformation capture (3C).** 3C assay was performed as described previously[47]. In brief, 5×10$^6$ cells were crosslinked with 1% formaldehyde for 10 min at room temperature. Crosslinked samples were digested overnight with 450 U of *Bgl*II (NEB) at 37 °C while shaking at 900 rpm. Diluted samples were ligated using T4 DNA ligase (Thermo) at 16 °C overnight. RNA was removed by

incubating samples with RNase A (Invitrogen) for 1 h at 37 °C. Cross-link was reversed by incubating aliquots with 10 mg ml$^{-1}$ proteinase K for 1 h at 50 °C and then overnight at 65 °C. DNA was purified with phenol/chloroform. The final quantitative PCR reactions were performed in triplicates by using SYBR Green with 100 ng DNA as templates. Primers for examining the enzymatic digestion efficiency and 3C-qPCR are listed in Supplementary Data 6.

**Apoptosis assay by flow cytometry.** Apoptosis assay was analyzed by using Annexin V-FITC kit according to the manufacturer's protocol (BD Biosciences, CatLog: 556547). In brief, cells were seeded in six-well plates and harvested at a density of 6 × 10$^5$ cells ml$^{-1}$, and then washed twice with ice-cold PBS. Cells were suspended with 100 μl 1× binding buffer containing 5 μl of Annexin V-FITC and 5 μl of propidium iodide (PI) and incubated for 10 min in dark at room temperature; apoptosis assay was analyzed by using a BD LSRFortessa Cell Analyzer (Becton-Dickinson, CA, USA).

**ChIP-seq library preparation and next-generation sequencing.** Adaptor oligonucleotides and primer sequences from Illumina were used for library construction and amplification. After PCR library amplification, size selection of adaptor-ligated DNA was performed using Agecourt AMPure XP Beads (Beckman Coulter). The AMPure XP Beads purified libraries were diluted and quantified by VAHTS$^{TM}$ Library Quantification Kit for Illumina. The libraries were denatured and diluted at a proper concentration for sequencing and finally sequenced on Illumina NextSeq500 (Guangzhou Institutes of Biomedicine and Health, Chinese Academy of Sciences) or HiSeq X-Ten (Annoroad Gene Technology Co., Ltd).

**RNA-seq analysis.** Raw reads were aligned to a Bowtie2 (v2.2.5)[48] indexed human genome (hg19 sourced from UCSC) using STAR (v2.5.2a)[49]. Gene expression levels were quantified as read counts generated by RSEM (v1.2.22)[50], with default settings. Raw tag counts were normalized for GC content using EDASeq (v2.8.0)[51]. Differential gene expression was called using DESeq2 (v1.10.1)[52]. Genes were considered as changing if their expressions were significantly different (*q* value < 0.05) and with above 1.5-fold-change, which were subsequently used in generating hierarchical clustering heatmaps.

**ChIP-seq analysis.** Before alignment, reads were qualified by FastQC (v0.11.2). Trim Galore (v0.4.4) was used to trim adaptor and low-quality reads if necessary. Trimmed reads were aligned to hg19 human genome assembly using Bowtie2 (v2.2.5)[48], with parameters --very-sensitive --end-to-end. Proper paired and high-quality mapped reads (MAPQ > 30) were selected with samtools (v1.2)[53]. PCR duplicates were removed by picard tools (v1.90), then reads were subjected to MACS2 (v2.1.0)[54]. For FLAG-tagged ChIP-seq, DNA from FLAG IP was used as the normalized control; peaks with *p* value less than 1e-4 were kept. For analyzing CTCF and RAD21 ChIP-seq data, peaks with *q* value less than 0.01 were kept for subsequent analysis. Signal tracks for each sample were generated using the MACS2 pileup function and were normalized to 1 million reads. Bigwig files were generated using the bedGraphToBigWig command for visualization. ChIP-seq heatmap and profiles were drawn with deeptools (v2.2.4)[55]. Peak annotation was performed using annotatePeaks function from homer[56]. Peak overlap was done using bedtools (v2.25.0)[57]. Differential peaks were called by MACS2 bdgdiff function with default parameters. De novo motif analysis of ChIP-seq peaks was performed using homer.

**GO analysis.** Gene ontology enrichment analyses were performed using GOseq (v1.17.4)[58]. *P* values were plotted to show the significance.

**ChIA-PET data analysis.** CTCF ChIA-PET data from HeLa-S3 cells were obtained from Rao et al.[7]. Raw reads were first trimmed by Trim Galore, then were analyzed by using ChIA-PET2 with default parameters for bridge linker ChIA-PET data[59].

**HiChIP data analysis.** HiChIP paired-end reads were first subjected to Trim Galore to remove adaptors. The trimmed reads were aligned to the hg19 genome using HiC-Pro software[60], with default settings except that reads were assigned to *Mbo*I restriction fragments. Valid reads generated by HiC-Pro were transformed to .hic format by juicer tools with KR normalization using hicpro2juicebox function. HiChIP interaction maps were visualized using Juicebox software[61], at 500, 25, 10 kb resolution in each analysis.

**HiChIP contact calling and differential loop analysis.** All HiChIP valid reads from HiC-Pro results were further processed to call loops using hichipper[62]. The overlapped CTCF peaks from our CTCF antibody-enriched ChIP-seq data were subjected to hichipper; only loops from these regions were called. Briefly, peaks were first extended 500 bp in either direction by default, and then merged if their distances were less than 500 bp. The restriction fragments containing these merged peaks were used as loop anchors. Loop distance less than 5 kb or larger than 2 Mb were further filtered as they were considered as either self-ligation or unlikely to be real loops. Significant interactions were called with mango pipeline[63], loops with

FDR < 0.05 were kept for each HiChIP data. Differential loops were identified using diffloop pipeline[64], with quickAssoc function, which was based on an over-dispersed Poisson regression model, then differential loops with high confidence were chosen as follows: first, loops with raw $p$ value < 0.05 were selected as primary differential loops. Then, in these primary differential loops, for FLAG-CTCF-s OE_vs._FLAG Control upregulated loops, the loops produced in the cells from two HiChIP replicates with FLAG-CTCF-s OE must not be zero. Similarly, for FLAG Control_vs._ FLAG-CTCF-s OE upregulated loops, the loops that either control HiChIP replicate is zero will be filtered. Finally, the filtered differential loops were considered as differential loops with high confidence and were used for further analysis.

**Reproducibility scatterplots and correlations**. HiChIP reproducibility analysis was performed as previously described[31], with some modifications. Briefly, HiChIP reproducibility scatterplots were generated by counting reads in all loop anchor regions, which were generated from our overlapped CTCF binding peaks. Raw reads from these anchor regions were normalized to 10 million filtered fragments, and then quantile normalized across all HiChIP experiments. The Pearson correlation between replicates was calculated using the cor() function in R.

**Statistical analysis and reproducibility**. The data are presented as mean ± s.d. as indicated in the figure legends. Two-tailed Student's $t$ test was used to assess statistical significance. A $p$ value < 0.05 was considered as statistically significant, *$p$ < 0.05, **$p$ < 0.01, ***$p$ < 0.001.

**Reporting Summary**. Further information on experimental design is available in the Nature Research Reporting Summary linked to this article.

## Data availability

Our sequencing data from ChIP-seq, RNA-seq, HiChIP have been deposited in the Gene Expression Omnibus and the accession number NCBI GEO: GSE108869. All other relevant data supporting the key findings of this study are available within the article and its Supplementary Information files or from the corresponding author upon reasonable request. The source data underlying Figs. 1c, 1e−h, 2j, 5b, 5h and 5j and Supplementary Figs 1b-f, 2a-b, 2d, 3, 5d, 10 are provided as a Source Data file. A reporting summary for this article is available as a Supplementary Information file.

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

## Acknowledgements

We thank Drs. Guohong Li, Wei Xie, Yuanchao Xue, and Xiaorong Zhang for helpful discussions. We thank Dr. Junjun Ding for providing pEF1α BirAV5-neo and pEF1α FLBIO-puro plasmid, and Dr. Guangjin Pan for providing pSin-FLAG vector. We thank Dr. Chunhui Hou for 3C suggestion. We thank Tian Zhang for providing uiPSC RNA samples for detecting the expression of CTCF-s isoform. This work was supported by the Strategic Priority Research Program of the Chinese Academy of Sciences (XDA16010502), the Ministry of Science and Technology of the People's Republic of China (2015CB964800, 2016YFA0100400, 2016YFA0100302), National Mega-project of China for Innovative Drugs (2018ZX09201002-005), National Natural Science Foundation of China (31471210, 31601050, 31850410463), Guangdong Natural Science Funds (2015A030308003), Science and Technology Planning Project of Guangdong Province of China (2016B030229006, 2017B030314056), Guangzhou Regenerative Medicine and Health Guangdong Laboratory (2018GZR110104007), Guangzhou Science Technology and Innovation Commission (201807010101, 201707020042), Guangdong Provincial Key Laboratory of Biocomputing (2016B030301007). The authors also gratefully thank the support from the Guangzhou Branch of the Supercomputing Center of the Chinese Academy of Sciences.

## Author contributions

H.Y. conceived, managed and arranged funding for the project; J.L. and K.H. performed most of the experiments and analyzed the data; G.H. and K.H. performed bioinformatics analysis; Y.L., X.D. and L.S. conducted the experiments; I.A.B., Y.-S.C., A.P.H., and Y.-G.Y. provided computational analysis; W.G. helped with FRAP experiment. J.W. provided the help for next-generation sequencing. Z.C. assisted the computational platform. H.Y., K.H., and J.L. wrote the manuscript. H.Y. approved the final version.

## Additional information

**Competing interests:** The authors declare no competing interests.

