## [Peer Review File · Nature Communications]

Reviewers' comments:

Reviewer #1 (Remarks to the Author):

The manuscript by Li et al reports characterization of the activities of a shorter non-canonical isoform of CTCF. My expertise is on alternative splicing, and I restrict most of my comments to that part of the manuscript (Figs 1 and 2).

The CTCFs isoform reported here is already well annotated e.g. Refseq (CTCT transcript variant 2), Ensembl (CTCF-202). Nevertheless, prior annotation does not diminish from the interest of defining novel functions of the CTCF-s isoform. In my opinion much of the data in Fig 2 addressing the mechanism of regulation of CTCF exons 3 and 4 is not strictly relevant to the remainder of the paper (notably the discussion does not mention anything about the RNA binding proteins that influence exon 3 & 4 splicing). This Figure seems to me to be of marginal importance for the major theme of the manuscript and it could be omitted without detriment to the remainder of the manuscript if any of my comments are difficult to address. While I have some reservations about the presentation of some of the data in Figs 1 and 2, this should not have a major influence on the decision whether to accept the manuscript for publication.

Specific comments

Major

1. The quantitation of relative amounts of CTCF and CTCFs is important (Fig 1g for protein and Supp 1e for mRNA), because if the CTCFs isoform is not abundant in any cells, then a lot of the subsequent experiments based upon ectopic overexpression are of more questionable significance. The western blot of Figure 1g appears to show that CTCFs is quite abundant in some cell types (predominant in H9). What is known about the specificity of the Ab used in Fig 1g? Is it equally reactive with both isoforms? However, the qRT-PCR appears to show that CTCF-s is a minor isoform in all tested cell types.

This quantitation of protein and mRNA is important and should be discussed explicitly in the main text (for the reason stated above). I would argue that Supp Fig 1e should be a main figure panel.

2. The CTCF-s specific knockdown (Fig 1c, f) using sh431 is not convincing. Statistical tests should be applied to the qRT-PCR data in Fig 1f. The western blot data (Fig 1e) needs some sort of quantitation, involving a serial dilution of the control sample to allow at least semi-quantitation of the knockdown efficiency. From visual inspection, the only shRNA that has a clear effect on protein levels is sh-1911. It is not necessarily surprising that the CTCF-s specific shRNA is not efficient given the very limited target encompassing the exon 2-5 spliced junction.

3. Why are separate panels used for the western blot in Fig 1e? Why is a single panel not shown for both isoforms, as in Fig 1g?

4. Page 5 and Figure 2b. The GFP minigenes are not adequately explained. The rationale is presumably that any exon included between the GFP exons will disrupt the GFP open reading frame, leading to loss of fluorescence. This sort of assay really needs validation first by RT-PCR to show that the fluorescence signal is explained by variations in exon inclusion. More information is needed on the CTCF regions included (e.g. what does "E3 only" mean) – simplest would be to provide genomic coordinates or the length of introns flanking each of the text exons.

5. Fig 2c,d. More information is needed on the precise regions of RNA used as bait in the pull down assay.

6. Fig 2h. What was the actual ratio of CTCF-s/CTCF in the reference condition (sh Ctrl)? It has arbitrarily been set to 1.0. However, if the starting absolute ratio was 0.01 (1%) the effects of hnRNPC depletion would represent an increase of CTCF-s from 1 to 3% of total CTCF. This might be statistically significant, but it would be of limited biological/mechanistic significance, and would call into question the claim that “these 9 RBPs are critical regulators...” (p7 line 3). On the other hand if the starting absolute ratio is 1.0, then the change would be from 50:50 to 75:25%. The y-axis here should show the absolute ratio or proportion of CTCF-s:CTCF.

7. Fig 2c. It is misleading to show a “spliceosome” assembling on the bait RNAs. As far as I can infer from the labels of the different RNAs (E3, E13 etc) none of the bait RNAs would be competent to fully assemble a spliceosome. As for the minigenes, the precise sequences or genome coordinates should be provided. The rationale for selecting these four regions is not well justified. Cis-regulatory elements that influence alternative exon inclusion/skipping can lie within exon bodies or within the flanking introns – typically up to 200 nt distant. Nevertheless, it has to be conceded that the approach has successfully identified proteins that influence CTCF exon 3 and 4 inclusion (subject to the caveat stated in the previous point).

Minor

1. P4, line 14-15. “lacks an N-terminus plus 2.5 zinc fingers”. Strictly speaking it lacks the sequences encoding 2.5 Zn fingers, but effectively the protein will lack three Zn finger domains.

2. P4 line 16 (Fig 1b). How was this nested PCR? What were the two pairs of primers used in the first and second rounds of PCR?

3. P5, line 13. “suggesting CTCF exon 3 and 4 could be alternatively spliced”. Might be better worded “...consistent with the fact that CTCF exons 3 and 4 are alternatively spliced.” Analysis of predicted splice site strengths is not needed to “suggest” that they are alternatively spliced – this has already been shown.

Reviewer #2 (Remarks to the Author):

In the present article, the authors tried to answer an interesting question about the potential role of alternative splicing of the genome organizer CTCF in genome organization and gene regulation and how it can affect cell behavior.

The paper is divided in two parts. In the first part, the authors identified a new isoform of CTCF, called CTCF-S. An alternative splicing of the exon 3 and 4 induces the production of a shorter protein, lacking the N-terminal domain and 2.5 zinc fingers out of 11. The existence of isoforms of genome organizers can be of course of broad interest and opens new horizons for regulation of genome expression and function. There are not many studies trying to answer this question. They further identified positive and negative regulators of this splicing event. Finally they showed that CTCF-S competes with CTCF for DNA binding, and resulting in decreased cohesin binding to the genome and altered CTCF/cohesin mediated long-range chromatin interactions.

In the second part, they tried to link CTCF-S and cell apoptosis. They showed that overexpression of CTCF-S results in increased cell apoptosis, increased phosphorylation of STAT1 and induction of Type 1 interferon pathway. They showed that CTCF-S disrupts a CTCF-binding site at a TAD boundary which results in enhancer-promoter interaction and up-regulation of the type I interferon gene IFI6. This up-regulation may then promote apoptosis.

The paper asks new question and is of broad interest in the fields of gene regulation and genome organization and function. However, it seems like the authors decided to put together two stories that are not equal in quality. Some questions should be answered, and maybe the second part (Fig 5, 6) should be reorganized (see comments below).

General comments:

- 1) The paper may need some language editing.
- 2) Half of the references are for the Material and Methods section. The authors could comment on other recent papers about CTCF, including the 2 papers on efficient CTCF inhibition through DEGRON and consequences on genome organization (Nora et al, 2017; Kubo et al, 2017).

Specific questions and comments:

1) Introduction:

- p3, line 4: "the mammalian genome is organized into structural and functional TADs". The "functional" significance of TADs seems to be still a debate depending on models, studies, etc.
- p3, line 9: there is not a single reference for the entire section on splicing.
- p3, line 16: "roles of canonical CTCF...", again the authors should cite articles.

2) Results first section: "Identification of new CTCF isoform in humans"

- The authors provide levels of CTCF or CTCF-S compared to control genes. But could the authors provide a clear comparison of the amount of the two isoforms with each other? At the mRNA and protein level? In other words what is the percentage of the CTCF and CTCF-S in the cells?
- Fig 1 e/f: could the authors provide a percentage of inhibition compared to control shRNA?
- Fig 1g: What is the third band at 70kDa?
- Fig 1: from the Fig 1g it seems that the used the Millipore (#07-729) CTCF- antibody for the WB. Is it the same for Fig 1e? This is also a general comment through the article; it is unclear which antibodies are used for WB, ChIP, etc? The authors should state it clearly in the results as well as in the Methods. Indeed, unless I missed a section, it seems that they only mention the 3 different antibodies in the Suppl Table 3 but they do not say which antibody was used for which experiment. Moreover, it would be useful to have a clear description of the different epitopes the different antibodies are recognizing. Finally, the authors could give details about their antibody.

3) Results second section: "CTCF is alternatively spliced through the action of specific RNA-binding proteins"

- Did the authors try to inhibit more than one factor at a time, to try to understand better the network of inhibition / activation of the CTCF splicing and see which ones have synergical effects, which ones are inhibiting each other etc?

4) Results third section: "CTCF-S competes with canonical CTCF long isoform binding in the genome"

- Same comment as in Fig 1, it is not clear which antibodies have been used and when.
- P7 line 15: As the FLAG ChIP-seq implies overexpression of CTCF and CTCF-S (endogenous gene + FLAG-gene), did the authors compare their FLAG- ChIP-seq of CTCF with normal CTCF-ChIP-seq? Do they find same peaks? Also it seems that overexpression of CTCF-S is much more important than CTCF (Suppl Fig 6a), could it influence the FLAG-ChIP-seq results?
- P8 line 12: I may have missed the information, but how did the authors perform the overexpression of CTCF-S. Is it the FLAG-gene? What is the level of overexpression?

5) Results fourth section: "CTCF-S competition alters genome-wide CTCF-mediated long-range chromatin contacts"

- Is the competition between CTCF and CTCF-S DNA binding affecting more certain types of loops / TADs / A-B compartments? In terms of size of loops / TADs, location within the nucleus (for example are they Lamina-Associating-Domains?), type of chromatin, cell cycle-related?

- Some interesting experiments (not required though) would be to inhibit specifically one of the 2 isoforms and analyze the consequences in terms of genome organization, as well as inhibiting one regulator after the other to see whether they affect different types of loops.

6) Results Fifth and sixth sections: CTCF-S and cell apoptosis

- In the figure 5h how significant is the percentage of the cells that promote apoptosis? In fig 5F by WB there is a big difference between the CTCF and CTCF-s promoting apoptosis. But in the fig 5h the double negative cells in both cases are <80%. Is there a big statistical difference that needs to be shown?

- The authors claim that CTCF-S has a role of cell apoptosis through phosphorylation of STAT1 and upregulation of the type 1 interferon signaling pathway. What are the mechanisms? Is it entirely due to the up-regulation of IFI6 through the CTCF-S mediated disruption of two TADs resulting in the interaction between IFI6 promoter and its distal enhancer?

- This is a nice example of consequences of competition between CTCF and CTCF-S that may have been inserted in the article the other way around, starting with the Fig 6 and disruption of the TAD boundary (which would have nicely followed Fig 4), and then going into the consequences on cell behavior with increase in cell apoptosis (Fig 5). Indeed a strong statement on a broad role for CTCF-S on cell apoptosis may have required more experiments: what are the mechanisms for cell apoptosis beside STAT1 phosphorylation?

7) Discussion:

The conclusion is very short and may include more discussion and hypotheses raised by the authors' findings. The authors could have discussed more recent studies on the role and function of CTCF and genome organizers. They could speculate more on the role of this new isoform, on how it is promoted or inhibited. Here are some questions for speculations/ discussions:

- What is the mechanism promoting the alternative splicing? When is alternative splicing promoted / inhibited? Could it be related to any particular physiological / pathological conditions?

- What are the benefits for the cells to have alternative isoform(s) of CTCF?

- What is the role of this isoform?

- Do you believe there might be other isoforms that can also compete with the canonical CTCF? (70kDa band?)

- Do you think this isoform exist in other species?

- How these alternatively spliced isoforms can affect chromatin and cellular activity?

Reviewer #3 (Remarks to the Author):

An alternative CTCF isoform antagonizes canonical CTCF occupancy and changes chromatin architecture to promote cell apoptosis.

Summary

The manuscript by Li et al, describes the existence of a CTCF splicing variant (CTCF-s, for CTCF-short isoform). They use biochemical RNA pull down assays to identify the proteins bound to canonical and short form isoforms. By ChIP-seq and Hi-ChIP, they show that CTCF-s competes with CTCF to regulate cohesin binding and mediate long-range contacts. This is by far the most interesting finding in the paper but the authors do not really explore the possible functional implications on gene regulation in any depth. One example illustrated in the paper is CTCF-s mediated activation of the IFI6 gene, but the association with the interferon signaling pathway is poorly explained. Overall, the findings described in the paper are promising, however the focus is very diffuse and there is no information about what regulates the expression of the short isoform. Furthermore, the paper needs to be written so that the reader does not have to go all over the place, shuttling between the text, methods and figure legends looking for information about an experiment. Right now it takes a lot of work to make sense of the data. Importantly, additional

experiments and analyses need to be performed to support the claims.

Major Comments

- "Due to the lack of a specific antibody against CTCF-s, we introduced a FLAG tagged CTCF and CTCF-s into HeLa-S3 cells, respectively." However, there is no information provided about how the transgene was introduced and whether there is one or more copy. It is also unclear if a single clone or a mixed population was used for the study.
- All the experiments are performed in the context of the presence of endogenous CTCF. A cleaner system would be required to support the statements made throughout the paper. Overexpression of CTCF-s is an artificial situation and it is not clear whether it competes with CTCF normally or only in a situation where it is in excess.
- Page 2: what is the physiological relevance of CTCF-s and what physiological conditions would promote CTCF-s expression? There is no discussion about this and no attempt to find out what regulates its expression.
- Expression of CTCF-s should be analyzed in a number of primary ex-vivo derived cell types. Also the authors need to check whether its expression is conserved in other animals.
- Supp fig 7a: As per the venn diagram CTCF-s and CTCF seems to bind to vastly differing binding sites. What could be the reason? The authors should comment on what endogenous CTCF could be doing to binding of both FLAG tagged long and short form isoforms.
- Page 4, related to figure 1e: judged by eye, the sh-431 does not really decrease the amount of CTCF-s protein. Quantification of the western blot and/or IF experiments are necessary to convince the reader of the efficiency of the depletion.
- Figure 1f: it should be indicated in the figure that it is expression relative to GAPDH. How long were the cells treated with shRNA before they observe depletion? It looks like expression of CTCF-s with sh431 decreases only about 30%. Perhaps that is why there is no clear difference by western blot. The format for the name of the sh should be consistent through the figures: is it sh-431 or sh431?
- Figure 1g: it is not clear whether the bands shown here are really specific to CTCF or CTCF-s? Showing that the band disappeared in a depleted or deleted condition is necessary.
- Page 8, related to figure 3: there is way less total CTCF peaks than normally seen in CTCF-ChIP-seq.
- Please describe the findings in the context of the crystal structures of CTCF ZFs and its contacts with DNA (Hashimoto et al, 2017 and Yin et al, 2017). In these studies, zinc fingers 1 and 2 do not make base specific contacts. It is unclear from the paper if ZF 3 is still intact or not in CTCF-s.
- In fig 3D and E, In the EMSA when CTCF is in excess, the entire DNA shifts up. However, when CTCF-s is in excess, there is no reduction in the signal of free probe. What is the probe that is used here – no information is provided? As per the ChIP-seq data, what is the efficiency of binding of CTCF and CTCF-s to the probe?
- Relative to figure 4: a co-IP of CTCF-s /CTCF and cohesin is required to test whether CTCF-s loses the ability to bind cohesin.
- In figure 4, Hi-ChIP should be performed with a FLAG antibody, comparing CTCF and CTCF-s overexpression. This will test whether CTCF-s on its own can make loops.
- Overall, the connection between CTCF/Cohesin peak changes and gene expression changes need further investigation. This is the most interesting finding so the authors need to explore whether there is differential gene expression related to loss of CTCF and cohesin binding. For example is CTCF-s overexpression linked to changes in loops and gene expression.
- Page 12: how do the authors explain that STAT1 phosphorylation was activated? They need to look at this in the context of chromatin bound phosphorylated STAT1 and a ChIP-seq of STAT1.
- Why does knocking down one gene, IFI6 impact apoptosis? Is this the only gene in the interferon pathway that regulates apoptosis?
- It is not clear how over-expression of CTCF-s leads to an inflammatory expression profile – is this true the other way round ie does inflammation (or treatment of cells with IFN γ) lead to over-expression of CTCF-s?
- Relative to figure 6: the choice of the example would be more robust if loop changes match with

CTCF/CTCF-s binding changes. From the figure 6d, the only peak that really looks different between CTCF and CTCF-s is not at a loop anchor. The authors should show IGV 1-dimensional track of the Hi-ChIP to compare with ChIP-seq peaks.

- It is unclear how well the IP part of the Hi-ChIP experiment worked. Screenshots of Juicebox showing loops should be included, not only the arcs.
- A screenshot of the raw interaction map at specific loci with differential loops should be shown in supplemental data to support the author's claims that the loops are differential.
- 4C-seq should be performed from different viewpoints across the region to validate their claim. 3C is very biased and does not give a full picture of interactions across the region. Furthermore, it appears that there are no controls for the 3C experiment shown.
- Overall, the discussion is superficial. There are major points not addressed in the paper that should be addressed in the conclusion as a perspective of future directions.

Minor Comments

- The culture conditions of all cell lines should be described in the Method section.
- All venn diagrams should be true to scale. For eg, in Supp fig. 7, 30817 seems bigger than 48822 by eye.
- The level of expression with respect to the transgene should be shown.
- Page 5: more information is required about the in silico prediction of splice sites.
- Supplementary figure 6d: the molecular weight is not shown. How do the authors explain that CTCF-s is more abundant than CTCF? What are all the bands for CTCF? The text in page 7 says that CTCF and CTCF-s are mainly located in the nucleus, but there is still a lot in the cytoplasm. For all Westerns the nucleus should be divided into 2 fractions: nucleoplasmic and chromatin-bound.
- Figure 3 and figure 4: it is not clear what the control situation is. Cluster 2 should be shown in figure 3g as a comparison.
- Figure 4e: how many total CTCF and CTCF-s loops?
- Figure 4i: what is the directionality of the motifs at CTCF loop anchors associated with peaks?
- Figure 5g: Does it mean that expressing CTCF exogenously leads to survival of cells over and above that of WT conditions? This doesn't really make sense.
- Page 12: what is the read depth for all the Hi-ChIP samples? A summary table of valid pairs of interactions and loops should be included containing all the information before and after any cutoff and processing step.
- Page 32: for analysis of differential Hi-ChIP, please explain why peaks were extended 500 bp and what the authors mean by restriction fragments containing the merged peaks that were used as loop anchors. Please include a schematic representation of how the analysis is performed.
- For the differential Hi-ChIP loops, does the p-value cutoff correspond to the raw or adjusted p-value.
- In Supplementary figure 6a, do the primers detect ectopic CTCF as well as the endogenous CTCF? A schematic representation of the primer location should be included. For CTCF, the level of CTCF-N and CTCF-C should be comparable as the primers for the full length CTCF have both ends of the protein.
- In the method section, the references of the antibodies used for ChIP and Hi-ChIP should appear.

Response to Referees letter

Reviewers' comments:

Reviewer #1 (Remarks to the Author):

The manuscript by Li et al reports characterization of the activities of a shorter non-canonical isoform of CTCF. My expertise is on alternative splicing, and I restrict most of my comments to that part of the manuscript (Figs 1 and 2).

The CTCFs isoform reported here is already well annotated e.g. Refseq (CTCT transcript variant 2), Ensembl (CTCF-202). Nevertheless, prior annotation does not diminish from the interest of defining novel functions of the CTCF-s isoform. In my opinion much of the data in Fig 2 addressing the mechanism of regulation of CTCF exons 3 and 4 is not strictly relevant to the remainder of the paper (notably the discussion does not mention anything about the RNA binding proteins that influence exon 3 & 4 splicing). This Figure seems to me to be of marginal importance for the major theme of the manuscript and it could be omitted without detriment to the remainder of the manuscript if any of my comments are difficult to address. While I have some reservations about the presentation of some of the data in Figs 1 and 2, this should not have a major influence on the decision whether to accept the manuscript for publication.

Reply: We appreciate the reviewer #1's comments about the alternative splicing part. Though CTCF-s is annotated, it has not been investigated and characterized by other groups. During the last five months, we have performed more experiments and have carefully revised our manuscript. So far, we believe we can address all the comments from reviewer #1 and we hope that our responses will satisfy the first reviewer.

Specific comments

Major

1. The quantitation of relative amounts of CTCF and CTCFs is important (Fig 1g for protein and Supp 1e for mRNA), because if the CTCFs isoform is not abundant in any cells, then a lot of the subsequent experiments based upon ectopic overexpression are of more questionable significance. The western

blot of Figure 1g appears to show that CTCFs is quite abundant in some cell types (predominant in H9). What is known about the specificity of the Ab used in Fig 1g? Is it equally reactive with both isoforms? However, the qRT-PCR appears to show that CTCF-s is a minor isoform in all tested cell types.

This quantitation of protein and mRNA is important and should be discussed explicitly in the main text (for the reason stated above). I would argue that Supp Fig 1e should be a main figure panel.

Reply: We thank the reviewer for their comments. In combination with comments from reviewer #3, we have expanded our tested cell lines and have also included *ex vivo* cultured PBMCs (Peripheral Blood Mononuclear Cells) as additional primary cells. For quantification at the mRNA level, as there are limited regions for designing TaqMan probes for CTCF-s, we used two different TaqMan probes to detect the expression of CTCF and CTCF-s, respectively. Therefore, we feel it is inappropriate to compare the levels across genes based on the RT-qPCR data. Instead, we direct the reviewer to the Western blot panels where we can detect both CTCF and CTCF-s using a single antibody (commercial anti-CTCF antibody indicated on the top, specifically recognized the C terminus of CTCF protein, the details are in **Supplementary Table 3**) on the same PVDF membrane (**Fig. 1f**). Based on the quantitation of these bands, we estimate that CTCF-s can occupy 0% - 40% of the total CTCF/CTCF-s protein complement. In addition, we have accepted reviewer #1's suggestion, and in this revised version of manuscript, we have moved Supplementary Fig. 1e into the main figure as **Fig. 1e**.

From the Western blot (Fig. 1f) results in combination with TaqMan results (Fig. 1e), we believe that CTCF-s is the minor isoform in all tested cell types.

Fig. 1f. Western blot of CTCF and CTCF-s in different human cell lines with anti-CTCF antibody (Millipore, 07-729). The locations of the full-length CTCF and CTCF-s were indicated.

2. The CTCF-s specific knockdown (Fig 1c, f) using sh431 is not convincing. Statistical tests should be applied to the qRT-PCR data in Fig 1f. The western

blot data (Fig 1e) needs some sort of quantitation, involving a serial dilution of the control sample to allow at least semi-quantitation of the knockdown efficiency. From visual inspection, the only shRNA that has a clear effect on protein levels is sh-1911. It is not necessarily surprising that the CTCF-s specific shRNA is not efficient given the very limited target encompassing the exon 2-5 spliced junction.

Reply: As suggested, we have added statistical test for the qRT-PCR data in **Fig. 1g** (originally Fig. 1f), and included a serial dilution of the control sample for the Western blot analysis (**Fig. 1h**, original Fig. 1e) and quantified the relative amount of CTCF-s and CTCF by ImageJ software. In addition, for improved understanding, we have replaced the names of original sh431, sh1264, sh1365, sh1812 and sh1911 with shCTCF-s#1, shCTCF#1, shCTCF#2, shCTCF-both#1 and shCTCF-both#2, respectively. To improve the knockdown efficiency of shCTCF-s#1 (originally sh431), in the new data, we have re-infected the shCTCF-s#1 stable cell lines with shCTCF-s#1 lentivirus particles and have detected the knockdown efficiency again. From the ImageJ quantification results, we indeed noticed that shCTCF-both#2 could clearly knock both CTCF and CTCF-s down, and shCTCF-s#1 could specifically knock CTCF-s down (0.46), but could not knock CTCF down (0.94).

Fig.1g &1h. (g) RT-qPCR analysis of two different CTCF isoforms after specific shRNA knockdown. (h) Western blot analysis of two different CTCF isoforms after specific shRNA knockdown.

3. Why are separate panels used for the western blot in Fig 1e? Why is a single panel not shown for both isoforms, as in Fig 1g?

Reply: In original Fig. 1g, we just wanted to confirm the existence of CTCF-s. However, in original Fig. 1e, we wanted to detect CTCF-s more accurately by using specific shRNA knockdowns. Since an unknown 70 kDa band, which is close to CTCF-s protein, influenced the detection accuracy of CTCF-s. Therefore, we cut the PVDF membrane in the middle to detect CTCF and CTCF-s, separately.

4. Page 5 and Figure 2b. The GFP minigenes are not adequately explained. The rationale is presumably that any exon included between the GFP exons will disrupt the GFP open reading frame, leading to loss of fluorescence. This sort of assay really needs validation first by RT-PCR to show that the fluorescence signal is explained by variations in exon inclusion. More information is needed on the CTCF regions included (e.g. what does “E3 only” mean) – simplest would be to provide genomic coordinates or the length of introns flanking each of the text exons.

[redacted]

[redacted]

5. Fig 2c,d. More information is needed on the precise regions of RNA used as bait in the pull down assay.

[redacted]

6. Fig 2h. What was the actual ratio of CTCF-s/CTCF in the reference condition (sh Ctrl)? It has arbitrarily been set to 1.0. However, if the starting absolute ratio was 0.01 (1%) the effects of hnRNPC depletion would represent an increase of CTCF-s from 1 to 3% of total CTCF. This might be statistically significant, but it would be of limited biological/mechanistic significance, and would call into question the claim that “these 9 RBPs are critical regulators...” (p7 line 3). On the other hand if the starting absolute ratio is 1.0, then the

change would be from 50:50 to 75:25%. The y-axis here should show the absolute ratio or proportion of CTCF-s:CTCF.

[redacted]

7. Fig 2c. It is misleading to show a “spliceosome” assembling on the bait RNAs. As far as I can infer from the labels of the different RNAs (E3, EI3 etc) none of the bait RNAs would be competent to fully assemble a spliceosome. As for the minigenes, the precise sequences or genome coordinates should be provided. The rationale for selecting these four regions is not well justified. Cis-regulatory elements that influence alternative exon inclusion/skipping can lie within exon bodies or within the flanking introns – typically up to 200 nt distant. Nevertheless, it has to be conceded that the approach has successfully identified proteins that influence CTCF exon 3 and 4 inclusion (subject to the caveat stated in the previous point).

[redacted]

Minor

1. P4, line 14-15. “lacks an N-terminus plus 2.5 zinc fingers”. Strictly speaking it lacks the sequences encoding 2.5 Zn fingers, but effectively the protein will lack three Zn finger domains.

Reply: We agree with reviewer #1’s comment. We have revised this sentence from “*This putative shorter isoform (we termed CTCF-s) lacks an N-terminal domain plus 2.5 zinc fingers (ZFs), but still contains 8 intact ZFs and full length*

C-terminal domain (Fig. 1a).” into “*This putative shorter isoform (we termed CTCF-s) lacks the sequences encoding N-terminus domain plus 2.5 zinc fingers (ZFs), but still effectively contains 8 intact ZFs and full length C-terminal domain (Fig. 1a).*” in the revised manuscript.

2. P4 line 16 (Fig 1b). How was this nested PCR? What were the two pairs of primers used in the first and second rounds of PCR?

Reply: We apologize for not providing the detailed information for nested PCR in the figure legend. For the first round of PCR, we have used F1(146-165) and R1(2652-2671) as the 1st set of primers which were marked black arrows in Fig. 1b. And for the second round of PCR, we have used F2(360-381) and F3(1024-1048) as forward primers, R2 (2506-2532) as reverse primer to amplify both isoform or long isoform only, respectively. To clarify the information clearly, we have added the detailed primer information into the revised figure legend.

3. P5, line 13. “suggesting CTCF exon 3 and 4 could be alternatively spliced”. Might be better worded “..consistent with the fact that CTCF exons 3 and 4 are alternatively spliced.” Analysis of predicted splice site strengths is not needed to “suggest” that they are alternatively spliced – this has already been shown.

Reply: We thank reviewer #1 for their comment. We have revised this sentence in this revised version of manuscript as suggested by reviewer #1.

Reviewer #2 (Remarks to the Author):

In the present article, the authors tried to answer an interesting question about the potential role of alternative splicing of the genome organizer CTCF in genome organization and gene regulation and how it can affect cell behavior.

The paper is divided in two parts. In the first part, the authors identified a new isoform of CTCF, called CTCF-S. An alternative splicing of the exon 3 and 4 induces the production of a shorter protein, lacking the N-terminal domain and 2.5 zinc fingers out of 11. The existence of isoforms of genome organizers can be of course of broad interest and opens new horizons for regulation of genome expression and function. There are not many studies trying to answer this question. They further identified positive and negative regulators of this splicing event. Finally they showed that CTCF-S competes with CTCF for DNA binding, and resulting in decreased cohesin binding to the genome and altered CTCF/cohesin mediated long-range chromatin interactions.

In the second part, they tried to link CTCF-S and cell apoptosis. They showed that overexpression of CTCF-S results in increased cell apoptosis, increased phosphorylation of STAT1 and induction of Type 1 interferon pathway. They showed that CTCF-S disrupts a CTCF-binding site at a TAD boundary which results in enhancer-promoter interaction and up-regulation of the type I interferon gene IFI6. This up-regulation may then promote apoptosis.

The paper asks new question and is of broad interest in the fields of gene regulation and genome organization and function. However, it seems like the authors decided to put together two stories that are not equal in quality. Some questions should be answered, and maybe the second part (Fig 5, 6) should be reorganized (see comments below).

Reply: We are grateful for the reviewer's positive comments on our work. As suggested, we have reorganized the second part (Fig. 5, 6) into a single figure as **Fig. 6** in the revised manuscript, and added a new **Fig. 5** into the manuscript to illustrate the connection between CTCF-s competition and transcriptional regulation.

General comments:

1) The paper may need some language editing.

Reply: We have sent our paper to some English native speakers for language editing, and we hope that the revised manuscript can meet the reviewer's expectations.

2) Half of the references are for the Material and Methods section. The authors could comment on other recent papers about CTCF, including the 2 papers on efficient CTCF inhibition through DEGRON and consequences on genome organization (Nora et al, 2017; Kubo et al, 2017).

Reply: For the clarity of data manipulation and for the reproducibility of our next generation sequencing data, we provided detailed data processing procedures, including all the software or packages used in this study, we have cited these necessary references in the Material and Methods section.

As suggested, we have introduced the papers about CTCF (Nora et al, Cell,169, 930-944(2017); Kubo et al, bioRxiv, 2017) as background in page 11 and cited the related references. We have further strengthened the introduction, particularly the part about alternative splicing. So far, we have included the paper on efficient CTCF inhibition through DEGRON and consequence on genome organization and transcriptional regulation. We have added a new figure as **Fig. 5** into the paper, which illustrates the relationship between CTCF-s competition and transcriptional regulation. And consistent with the Nora et al., (2017) paper, we have added the following sentences into the revised manuscript "***Upon CTCF-s gain of function, 130 genes were up-regulated and 111 genes were down-regulated (Fig. 5a). Integration with CTCF ChIP-seq data revealed that nearly 46% (51/111) of the downregulated genes had CTCF bound within the promoter, as opposed to 23.8% (30/130) of the upregulated genes (Fig. 5b). Moreover, consistent with a recent report about the link between auxin induced degradation of CTCF and transcriptional regulation (Nora et al, Cell,169, 930-944(2017)), of those downregulated genes, we also observed significantly decreased CTCF binding at promoter regions (Fig. 5c). While of those upregulated genes, we noticed that the distance between promoter and flanking enhancer was much closer (Fig. 5d)***".

We have also added two references about the crystal structure of CTCF-ZFs into the manuscript (Hashimoto et al, Mol Cell 66,711-720(2017); Yin et al., Cell Res 27, 1365-1377(2017)), which specified the interaction between CTCF ZFs and the specific nucleotides.

We have also strengthened our discussion. For example, we added a longer discussion on the canonical long isoforms of CTCF, and its reported new biological functions. After those improvements, we have increased the reference number to 43 for the main text and 22 for the method section in this revised manuscript.

Specific questions and comments:

1) Introduction:

- p3, line 4: “the mammalian genome is organized into structural and functional TADs”. The “functional” significance of TADs seems to be still a debate depending on models, studies, etc.

Reply: We agree with reviewer #2's comment, and have removed the word ‘functional’ in the revised manuscript.

- p3, line 9: there is not a single reference for the entire section on splicing.

Reply: We have revised this paragraph, and have added 7 references to this section. **“Alternative splicing, is the process by which splice sites in primary transcripts are differentially selected to produce structurally and functionally distinct mRNA and protein isoforms (Matlin et al., Nat Rev Mol Cell Biol 6, 386-398(2005)). It provides a powerful mechanism to expand the functional and regulatory capacity of metazoan genomes. Genome-wide studies estimated that 90-95% of human genes undergo alternative splicing (Pan et al., Nat Genet 40, 1413-1415(2008)), and a subset of alternative splicing events have been identified that are regulating development (Wang et al, Nature 456,470-476(2008); Zhang et al., Cell 166, 1147-1162 (2016)), tissue identity (Baralle et al., Nat Rev Mol Cell Biol 18, 437-451(2017)), pluripotency (Gabut et al., Cell 147,132-146(2011)), and tumor proliferation (Qi et al., Nat Commun 7(2016)). Yet, the role of alternative splicing in chromatin organization**

has not been widely explored, yet it may be an important factor considering the widespread splicing of human mRNAs, as it may control chromatin architecture to modulate regulatory pathways that can affect cell fate or function.”

- p3, line 16: “roles of canonical CTCF...”, again the authors should cite articles.

Reply: We have cited 2 references for this sentence in the revised manuscript, one is a review paper published in Cell in 2009 (Phillips JE, Corces VG, Cell, 2009), and another is a recent review (Ghirlando R, Felsenfeld G, Genes & Development, 2016) about CTCF in higher-order chromatin organization.

2) Results first section: “Identification of new CTCF isoform in humans”

- The authors provide levels of CTCF or CTCF-S compared to control genes. But could the authors provide a clear comparison of the amount of the two isoforms with each other? At the mRNA and protein level? In other words what is the percentage of the CTCF and CTCF-S in the cells?

Reply: We used two different TaqMan probes and the corresponding primers to detect the RNA levels of CTCF and CTCF-s, respectively. As the different sets of primers might have different amplification efficiencies, therefore, we feel it is inaccurate to cross-compare CTCF and CTCF-s using RT-qPCR. Instead, we direct the reviewer to the Western blots, which compare the two isoforms at the protein level. And as shown in **Fig. 1f**, when we quantified the CTCF-s/CTCF at protein level, we found that CTCF-s accounts for 0% to 43% (averaging about 10%) among the 16 cell lines we tested.

Fig.1f. Western blot of CTCF and CTCF-s in different human cell lines. The locations of the full-length CTCF and CTCF-s are indicated.

- Fig 1 e/f: could the authors provide a percentage of inhibition compared to control shRNA?

Reply: We would like to thank reviewer #2 for their suggestion. In the revised version we have provided the percentage of inhibition (expressed as a fold-change), compared to control shRNA, as requested.

Fig.1g. RT-qPCR analysis of two different CTCF isoforms after specific shRNA knockdown. The data are reported as mean values \pm s.d. with the indicated significance by using a two-tailed student's *t* test (* p <0.05, ** p <0.01, *** p <0.001).

- Fig 1g: What is the third band at 70kDa?

Reply: We are unclear what this band is. It may be a non-specific band or a degraded product of full-length CTCF.

- Fig 1: from the Fig 1g it seems that the used the Millipore (#07-729) CTCF-antibody for the WB. Is it the same for Fig 1e? This is also a general comment through the article; it is unclear which antibodies are used for WB, ChIP, etc? The authors should state it clearly in the results as well as in the Methods. Indeed, unless I missed a section, it seems that they only mention the 3 different antibodies in the Suppl Table 3 but they do not say which antibody was used for which experiment. Moreover, it would be useful to have a clear description of the different epitopes the different antibodies are recognizing. Finally, the authors could give details about their antibody.

Reply: We thank reviewer #2 for their comment. In this revised manuscript, we followed reviewer #2's suggestion and clearly stated which antibody was used for which experiments. Specifically, we used anti-CTCF-antibody from Millipore (#07-729) for the Western blot in **Fig.1f** (originally Fig. 1g), which recognizes the C terminal of CTCF. For the Western blot in **Fig. 1h** (originally Fig. 1e), we also used this antibody for detection of the long isoform CTCF. And we used anti-CTCF antibody raised by our laboratory for detecting CTCF-s, which recognizes the whole C terminal of CTCF protein. For CTCF ChIP-seq, we used anti-CTCF antibody from Active Motif (CatLog: 61311), which was raised only against a peptide within the N terminal of human CTCF

and could not recognize CTCF-s (lack N terminal and 3 ZFs), and therefore it could be used to detect the long canonical isoform of CTCF only. We have added this detailed information into the Figures, Methods, as well as **Supplementary Table 3**.

3) Results second section: “CTCF is alternatively spliced through the action of specific RNA-binding proteins”

- Did the authors try to inhibit more than one factor at a time, to try to understand better the network of inhibition / activation of the CTCF splicing and see which ones have synergical effects, which ones are inhibiting each other etc?

[redacted]

4) Results third section: “CTCF-S competes with canonical CTCF long isoform binding in the genome”

- Same comment as in Fig 1, it is not clear which antibodies have been used and when.

Reply: For this section, we performed ChIP-seq using an anti-CTCF antibody that was purchased from Active Motif (CatLog: 61311, Rabbit polyclonal antibody), which was raised against a peptide within the N-terminal region of human CTCF. We have provided the detailed information into **Supplementary Table 3**.

- P7 line 15: As the FLAG ChIP-seq implies overexpression of CTCF and CTCF-S (endogenous gene + FLAG-gene), did the authors compare their FLAG- ChIP-seq of CTCF with normal CTCF-ChIP-seq? Do they find same peaks? Also it seems that overexpression of CTCF-S is much more important than CTCF (Suppl Fig 6a), could it influence the FLAG-ChIP-seq results?

Reply: We compared FLAG ChIP-seq for FLAG-CTCF with normal CTCF ChIP-seq, and we found nearly 81% (16,143/20,010) of FLAG-ChIP-seq peaks of CTCF overlapped with normal CTCF ChIP-seq peaks from ENCODE (GSE33213) (**Supplementary Fig. 6e**). To further confirm this result, we also performed ChIP-seq for a biotin tagged CTCF, and compared the biotin-enriched CTCF peaks with normal CTCF ChIP-seq. We found that the biotin ChIP-seq data has more CTCF peaks than that of the FLAG tag, and 86.86% (51,077/58,806) of the ENCODE CTCF peaks overlapped with biotin-enriched CTCF peaks, indicating that biotin tag is much more effective than the FLAG tag. In the revised manuscript, we have replaced the original FLAG ChIP-seq data with biotin ChIP-seq results.

Fig. 3a and 3b. (a) Venn diagram showing the overlapping of biotin-enriched CTCF or CTCF-s peaks with a normal CTCF ChIP-seq from ENCODE (GSE33213). (b) Genomic tracks showing CTCF peaks from ENCODE (GSE33121), or biotin enriched CTCF or CTCF-s peaks at chromosome 12: 121,055,606-121,842,881 (785 kb).

- P8 line 12: I may have missed the information, but how did the authors perform the overexpression of CTCF-S. Is it the FLAG-gene? What is the level of overexpression?

Reply: Yes, both CTCF and CTCF-s were FLAG-tagged. We generated both FLAG-CTCF and FLAG-CTCF-s stable cell lines, and tested the level of both mRNA and protein levels for both CTCF and CTCF-s, respectively (**Supplementary Fig. 6**). We used two sets of primers (CTCF-N and CTCF-C) which target the N terminal and C terminal of CTCF, respectively, to detect the overexpression of CTCF and CTCF-s at the mRNA level (**Supplementary Fig. 6a**) and we further used anti-FLAG antibody to examine the overexpression at the protein level (**Supplementary Fig. 6b**). From **Supplementary Fig. 6**, we noticed that both FLAG-CTCF and FLAG-CTCF-s were successfully overexpressed.

5) Results fourth section: “CTCF-S competition alters genome-wide CTCF-mediated long-range chromatin contacts”

- Is the competition between CTCF and CTCF-S DNA binding affecting more certain types of loops / TADs / A-B compartments? In terms of size of loops / TADs, location within the nucleus (for example are they Lamina-Associating-Domains?), type of chromatin, cell cycle-related?

[redacted]

[redacted]

- Some interesting experiments (not required though) would be to inhibit specifically one of the 2 isoforms and analyze the consequences in terms of genome organization, as well as inhibiting one regulator after the other to see whether they affect different types of loops.

Reply: We would like to thank reviewer #2 for suggesting new experiments. We agree that it would be interesting to analyze the consequence of higher-order genome organization after inhibiting one of the two isoforms. We have put this suggestion into the discussion, and we are planning to perform those and other experiments in future studies.

6) Results Fifth and sixth sections: CTCF-S and cell apoptosis

- In the figure 5h how significant is the percentage of the cells that promote apoptosis? In fig 5F by WB there is a big difference between the CTCF and CTCF-s promoting apoptosis. But in the fig 5h the double negative cells in both cases are <80%. Is there a big statistical difference that needs to be shown?

Reply: In combination with the comments from reviewer #3, and due to poor results for STAT1 and pSTAT1 ChIP-seq, we could not provide STAT1 and pSTAT1 ChIP-seq to mechanistically illustrate the connection between CTCF-s on pSTAT1 activation and apoptosis, therefore we have removed the Western blot results (original Fig.5f).

For statistical significance, as suggested, we have added the mean \pm SD for each quarter and added a p value to show their statistical significance in the main text of the revised manuscript. For example, ***“Our results further showed that rescue of CTCF-s can promote cell apoptosis (9.95 ± 0.7 v.s. 6.89 ± 0.27 , $p = 0.0021$) while rescue of CTCF did not affect apoptosis (7.08 ± 0.14 v.s. 6.89 ± 0.27 , $p = 0.34$) (Fig. 6d).”*** and ***“Our results showed that apoptosis induced by CTCF-s was inhibited by IFI6 silencing even after DNA damage (54.25 ± 4.88 v.s. 41.9 ± 2.26 , $p = 0.083$) (Fig. 6e), suggesting that IFI6-type I interferon-signaling pathway could be an important pathway for CTCF-s-mediated apoptosis.”***

- The authors claim that CTCF-S has a role of cell apoptosis through phosphorylation of STAT1 and upregulation of the type 1 interferon signaling pathway. What are the mechanisms? Is it entirely due to the up-regulation of IFI6 through the CTCF-S mediated disruption of two TADs resulting in the interaction between IFI6 promoter and its distal enhancer?

Reply: In **Fig. 6e**, we have observed that overexpression of CTCF-s affected apoptosis-related genes and induced cell apoptosis, while knocking-down IFI6, CTCF-s could no longer induce apoptosis. Therefore, we concluded that CTCF-s promotes apoptosis mainly through IFI6. In this study, we discovered that CTCF-s competes CTCF binding and disrupts CTCF-loops at IFI6 locus. Indeed, CTCF-s gain significantly enhances IFI6 enhancer-promoter interaction. Mechanistically speaking, this might be an important way to modulate IFI6 gene expression and cell apoptosis.

- This is a nice example of consequences of competition between CTCF and CTCF-S that may have been inserted in the article the other way around, starting with the Fig 6 and disruption of the TAD boundary (which would have nicely followed Fig 4), and then going into the consequences on cell behavior with increase in cell apoptosis (Fig 5). Indeed a strong statement on a broad role for CTCF-S on cell apoptosis may have required more experiments: what are the mechanisms for cell apoptosis beside STAT1 phosphorylation?

Reply: In the revised manuscript, we have added a new **Fig. 5**, which proposed that the loss of CTCF and cohesin binding caused by CTCF-s competition has a relationship with transcriptional regulation, and then analyzed the consequences of biological functions caused by CTCF-s gain of function (**Fig. 6**). We proposed that up-regulation of IFI6 was caused by disruption of CTCF-mediated chromatin loops and abnormal E-P interaction through CTCF-s gain of function.

Fig. 5. CTCF-s competition and transcriptional regulation. (a) Hierarchical clustering of differentially expressed (DE) gene profiles after overexpression of FLAG-CTCF-s in

HeLa-S3 cells. Fold-change was relative to the mean of the FLAG control. **(b)** Analysis of CTCF binding at promoters of DE genes, from panel a. **(c)** CTCF-s down-regulated genes tended to decrease CTCF binding at their promoter regions. **(d)** Up-regulated genes by CTCF-s tended to have enhancers that were closer to the TSS than down-regulated genes.

7) Discussion:

The conclusion is very short and may include more discussion and hypotheses raised by the authors' findings. The authors could have discussed more recent studies on the role and function of CTCF and genome organizers. They could speculate more on the role of this new isoform, on how it is promoted or inhibited. Here are some questions for speculations/ discussions:

- What is the mechanism promoting the alternative splicing? When is alternative splicing promoted / inhibited? Could it be related to any particular physiological / pathological conditions?
- What are the benefits for the cells to have alternative isoform(s) of CTCF?
- What is the role of this isoform?
- Do you believe there might be other isoforms that can also compete with the canonical CTCF? (70kDa band?)
- Do you think this isoform exist in other species?
- How these alternatively spliced isoforms can affect chromatin and cellular activity?

Reply: We appreciate the reviewer #2's careful comments on our discussion. We have revised our discussion section in the manuscript as described below. We have discussed balance control of RBPs on gene alternative splicing and cell fate determination and added the following sentences in the revised manuscript as below "***In this study, we identified an alternatively spliced CTCF-s short isoform and identified 9 RBP candidates that positively or negatively control CTCF alternative splicing through screening. The balance of these splicing regulators precisely controls the splicing ratios of specific genes in the living compartments (Fu & Ares, Nat Rev Genet 15,689-701(2014)), and imbalance of splicing factors may contribute to biological and/or developmental abnormality or carcinogenesis (Dvinge***

et al., Nat Rev Can 16, 413-430(2016)). Therefore, the properly regulation of CTCF alternative splicing by these 9 newly identified splicing factors could be important for CTCF functions in regulating gene expression and cell fate determination.”

We have also discussed DNA binding motif and competition between CTCF and CTCF-s and added the following sentences into the discussion **“Two recent investigations about the crystal structure of CTCF ZFs specify nucleotides bound by each ZFs (Hashimoto *et al., Mol Cell* 66,711-720(2017); Yin *et al., Cell Res* 27, 1365-1377(2017)). Consistently, biotin-CTCF specifically binds the 15 bp core DNA motif, and biotin-CTCF-s, which lacks 7 aa (HKCPDCD) of the 24-aa ZF3, no longer recognizes the 2 core motif (C/G T/C) for ZF3 and only reserves G/A with decreased specificity. Our data suggested that CTCF-s competes with CTCF to reduce cohesin binding, and further disrupt chromatin looping mediated by CTCF, providing a new eviction mechanism for CTCF-mediated chromatin looping that is not triggered by mutations of CTCF binding sites or alteration of CTCF binding orientation (Tang *et al., Cell* 163,1611-1627(2015); Guo *et al., Cell* 162,900-910(2015)). It was suggested that recruitment of cohesin to most CTCF binding sites is in a CTCF-dependent manner (Parelho *et al., Cell* 132,422-433(2008); Wendt *et al., Nature* 451,796-801(2008)). Consequently, the reduction of cohesin could be explained if CTCF-s is impaired to recruit CTCF/cohesin, and so competes away CTCF and cohesin. Indeed, the interaction of CTCF and cohesin seems much stronger than that of CTCF-s and cohesin (Supplementary Fig. 9d), which might be another reason to characterize the decreases of cohesin- and CTCF-binding in their common targets and to explain why CTCF-s gain-of-function changes CTCF-mediated chromatin looping in the genome.”**

We have also added CTCF-s specific bindings, CTCF splicing pattern and future perspectives into the discussion as below **“In this study we have mainly focused on the roles of CTCF-s in the context of canonical CTCF binding, however, there was 12,644 CTCF-s binding sites that did not overlap with CTCF peaks (Fig. 3a). This intriguing observation suggests**

that in addition to its competitive function, CTCF-s binds to genomic regions independently of CTCF and provides an additional layer of genome organization. Further, the splicing pattern of CTCF varies in different cell types, providing a diversity of CTCF-mediated higher-order chromatin organization possibilities in different cell types. Therefore, the investigation of CTCF alternative splicing mechanisms and characterization of the biological or pathological functions, especially in cancer, of this newly reported CTCF short isoform will need to be addressed in the future.”

Finally, we have given a brief summary about this study as below. ***“In summary, we conclude that ectopic expression of CTCF-s can disrupt the chromatin architecture maintained by CTCF. Therefore, CTCF-s may act as a modulator or ‘fine-tuner’ of CTCF activity, balancing CTCF-mediated chromatin organization and transcriptional regulation. The competition between CTCF and CTCF-s ultimately leads to an effect on cell phenotype as the cells activate an apoptotic pathway. Our study demonstrates the importance of alternatively spliced isoforms in chromatin and cellular activity.”***

We provided a clearer statement of the novelty and significance of this study and a spotlight for further investigation and we hope that this discussion could meet the reviewer’s expectation and readers’ interest.

Reviewer #3 (Remarks to the Author):

An alternative CTCF isoform antagonizes canonical CTCF occupancy and changes chromatin architecture to promote cell apoptosis.

Summary

The manuscript by Li et al, describes the existence of a CTCF splicing variant (CTCF-s, for CTCF-short isoform). They use biochemical RNA pull down assays to identify the proteins bound to canonical and short form isoforms. By ChIP-seq and Hi-ChIP, they show that CTCF-s competes with CTCF to regulate cohesin binding and mediate long-range contacts. This is by far the most interesting finding in the paper but the authors do not really explore the possible functional implications on gene regulation in any depth. One example illustrated in the paper is CTCF-s mediated activation of the IFI6 gene, but the association with the interferon signaling pathway is poorly explained. Overall, the findings described in the paper are promising, however the focus is very diffuse and there is no information about what regulates the expression of the short isoform. Furthermore, the paper needs to be written so that the reader does not have to go all over the place, shuttling between the text, methods and figure legends looking for information about an experiment. Right now it takes a lot of work to make sense of the data. Importantly, additional experiments and analyses need to be performed to support the claims.

Reply: We appreciate reviewer #3's positive comments on our study. Following reviewer #3's suggestion, we have extended and performed more work. With these new data, we have added a new figure (**Fig. 5**) into the manuscript in which we describe the connection between CTCF-s competition and transcriptional regulation on pages 13-14. We have also revised the manuscript, paying particular focus to the logical flow. We have included more data for CTCF and CTCF-s from biotin-tagged ChIP-seq, revising the Figures and Figure legends to make the Figures easier to understand and strengthening the discussion, and providing more detailed information in the method section and Supplementary Tables. Moreover, we have removed the Western blot result for STAT1 and pSTAT1, and now focus on the effect of

alteration of higher-order chromatin organization triggered by CTCF/CTCF-s competition on IFI6 activation.

We hope that our responses to the comments raised by reviewer #3 can meet their expectations.

Major Comments

- "Due to the lack of a specific antibody against CTCF-s, we introduced a FLAG tagged CTCF and CTCF-s into HeLa-S3 cells, respectively." However, there is no information provided about how the transgene was introduced and whether there is one or more copy. It is also unclear if a single clone or a mixed population was used for the study.

Reply: We thank the reviewer #3 for their comment. We cloned the CDSs for both CTCF and CTCF-s into pSin-3 × FLAG Vector with double enzyme digestion, respectively. We packaged pSin-3 × FLAG-CTCF or pSin-3 × FLAG-CTCF-s lentivirus in 293T cells, and then harvested and filtered lentivirus. Filtered lentivirus were used to infect HeLa-S3 cells for 24 hr, and then screened with 4 µg/ml puromycin for several days. Finally, we selected both FLAG-CTCF and FLAG-CTCF-s overexpressing stable cell lines. We have added detailed information to the method section of our revised manuscript.

- All the experiments are performed in the context of the presence of endogenous CTCF. A cleaner system would be required to support the statements made throughout the paper. Overexpression of CTCF-s is an artificial situation and it is not clear whether it competes with CTCF normally or only in a situation where it is in excess.

Reply: This is an insightful comment from the reviewer, and we have tried to generate a cleaner system to study the relationship between CTCF/CTCF-s and DNA binding by knocking CTCF or CTCF-s out. However, it was impossible for us to design a strategy to knock CTCF-s out as ultimately CTCF-s is an exon skipping event, and if we knocked CTCF-s out, then both CTCF-s and CTCF would be deleted in cells at the same time. Importantly, when we tried to delete the CTCF full length, by targeting the N terminus of

CTCF, the cells would gradually become senescent and die. We do not think that a knockout strategy will be successful. As an Alternative, to demonstrate that CTCF-s competes with CTCF *in vivo*, we performed an *in vitro* EMSA experiment (**Fig. 3d**), we noticed that increasing the amount of CTCF could lead to the binding reduction of CTCF-s onto the CTCF binding motif. Vice versa, our data indicated that the increasing amount of CTCF-s could result in the binding reduction of CTCF onto its binding motif as well. These data support the idea that CTCF and CTCF-s compete.

- Page 2: what is the physiological relevance of CTCF-s and what physiological conditions would promote CTCF-s expression? There is no discussion about this and no attempt to find out what regulates its expression.

Reply: We thank for the reviewer #3's comment. In the revised manuscript, we have analyzed the GTEx data (normal tissues) (Nat Genet 45,580-585(2013)) and TCGA datasets (cancerous tissues) (Weinstein et al., Nat Genet 45,1113-1120(2013)) for the expression of CTCF and CTCF-s. Surprisingly CTCF-s expression was widespread in many tissues, and importantly the CTCF-s/CTCF ratio was more frequently altered in cancerous tissues.

In addition, we have added the following sentence into the discussion in page 18. ***“Therefore, mechanistic investigation of the transcriptional circuit of CTCF-s and characterization of the biological or pathological functions, especially in cancer, of this newly reported CTCF-s short isoform will need to be addressed in the future.”***

- Expression of CTCF-s should be analyzed in a number of primary ex-vivo derived cell types. Also the authors need to check whether its expression is conserved in other animals.

Reply: This is an excellent suggestion by the reviewer. We have added Western blot data for ex-vivo derived PBMCs, and have expanded the tested cancer cell lines by Western blot. We found that CTCF-s expression accounts for 0% to 43% among 16 cell lines we tested. For expression in other species, we looked in the NCBI databases, but, to our knowledge, CTCF-s is a

human-specific transcript. Therefore, we only focused on studying the expression level of CTCF-s in human cell lines.

- Supp fig 7a: As per the venn diagram CTCF-s and CTCF seems to bind to vastly differing binding sites. What could be the reason? The authors should comment on what endogenous CTCF could be doing to binding of both FLAG tagged long and short form isoforms.

Reply: In the first version, due to lack of specific antibody recognizing CTCF-s, to separate the binding sites between CTCF and CTCFs, we generated both FLAG-CTCF and FLAG-CTCF-s stable cell lines and performed FLAG ChIP-seq for FLAG-CTCF and FLAG-CTCF-s, respectively. Though the enriched CTCF/CTCF-s peaks contained the CTCF core motifs, our data indicated that there was less CTCF binding sites identified using anti-FLAG antibody, suggesting that FLAG ChIP efficiency is less sensitive than normal CTCF ChIP-seq. To resolve this problem, we performed ChIP-seq using several different tags. Fortunately, we found that the biotin-tag performed well, in both the biotin-CTCF and biotin-CTCF-s ChIP-seq experiments. Our data indicated that biotin ChIP-seq data for biotin-CTCF got 72,937 CTCF peaks, and 86.86% (51,077/58,806) of CTCF peaks from ENCODE overlapped with our biotin-enriched CTCF peaks, which proved to be a more sensitive and effective method for ChIP-seq analysis. Hence, we also compared biotin-enriched CTCF-s peaks with CTCF, and 68.8% (28,459/41,362) of CTCF-s peaks overlapped with CTCF peaks, which was more reasonable. Therefore, we have replaced the FLAG ChIP-seq data with the new biotin ChIP-seq data.

Fig. 3a. Venn diagram showing the overlapping of biotin-enriched CTCF or CTCF-s peaks with a normal CTCF ChIP-seq from ENCODE (GSE33213).

- Page 4, related to figure 1e: judged by eye, the sh-431 does not really decrease the amount of CTCF-s protein. Quantification of the western blot

and/or IF experiments are necessary to convince the reader of the efficiency of the depletion.

Reply: To confirm this result, we have repeated these experiments several times by adding a dilution of control sample and strengthened the knockdown efficiency of shCTCF-s#1 (original sh431) and by performing two rounds of lentivirus infection. Our results indeed showed that shCTCF-s#1 (originally sh431) could decrease the amount of CTCF-s protein without affecting the protein level of CTCF (**Fig. 1h**).

- Figure 1f: it should be indicated in the figure that it is expression relative to GAPDH. How long were the cells treated with shRNA before they observe depletion? It looks like expression of CTCF-s with sh431 decreases only about 30%. Perhaps that is why there is no clear difference by western blot. The format for the name of the sh should be consistent through the figures: is it sh-431 or sh431?

Reply: We had indicated the expression relative to GAPDH in the original figure 1f, however, the first reviewer requested us to express as a percentage of inhibition. Therefore, we are wondering whether this is okay with the third reviewer (the data is now in **Fig. 1g**). We generated stable cell lines that can maintain shRNA targeting CTCF-s for several passages. To obtain these stable cell lines, we packaged the shRNA lentiviruses in 293T cells and then infected HeLa-S3 cells for 48 hr, and screened with puromycin for several days to obtain stable cell lines. To increase reader clarity, we have replaced the original names for the shRNA oligos from sh431, sh1264, sh1365, sh1812 and sh1911 to shCTCF-s#1, shCTCF#1, shCTCF#2, shCTCF-both#1 and shCTCF-both#2, respectively. To improve the knockdown efficiency of shCTCF-s#1 (originally sh431), in the new manuscript, we re-infected the shCTCF-s#1 stable cell lines with shCTCF-s#1 lentivirus particles and detected the knockdown efficiency again, and performed two-tailed Student's t test and added “*” showing their statistical significance. We also repeated these experiments and used Western blot to show the levels of CTCF and CTCF-s. From the ImageJ quantification results, shCTCF-s#1 could

specifically knock CTCF-s down (0.46), but could not knock CTCF down (0.94).

Fig.1g &1h. (g) RT-qPCR analysis of two different CTCF isoforms after specific shRNA knockdown. **(h)** Western blot analysis of two different CTCF isoforms after specific shRNA knockdown.

• Figure 1g: it is not clear whether the bands shown here are really specific to CTCF or CTCF-s? Showing that the band disappeared in a depleted or deleted condition is necessary.

Reply: In the first version of our manuscript, we have already performed isoform-specific shRNA knockdown and shown knockdown efficiencies for both CTCF and CTCF-s in the original Fig. 1e. In fig.1e, sh431 (now shCTCF-s#1) targeted to the exon2-5 spliced junction region, and can specifically knock down CTCF-s; sh1264 (shCTCF#1) and sh1365 (shCTCF#2) targeted the exon 3 and 4 specifically knocking down CTCF; sh1812 (shCTCF-both#1) and sh1911 (shCTCF-both#2) targeted to C-terminal regions of CTCF and CTCF-s. In the revised manuscript, we have included a serial dilution of the control sample for the Western blot analysis (**Fig. 1h**, originally Fig. 1e) and quantified the relative amount of CTCF-s and CTCF by ImageJ software. From the ImageJ quantification, we indeed noticed that shCTCF-both#2 could clearly knock down both CTCF and CTCF-s, while shCTCF#1 and shCTCF#2 could partially knock down CTCF but could not knock CTCF-s down at all, and shCTCF-s#1 could specifically knock CTCF-s down (0.46), but could not knock CTCF down (0.94). The isoform specific knockdown experiments suggest that the bands are specific for CTCF and CTCF-s.

- Page 8, related to figure 3: there is way less total CTCF peaks than normally seen in CTCF-ChIP-seq.

Reply: Due to the poor efficiency of the FLAG ChIP-seq, we indeed got less CTCF peaks from FLAG ChIP-seq than normal CTCF ChIP-seq. In the revised manuscript, as described above, we selected a more effective biotin tag for the ChIP experiments, and we have replaced the FLAG ChIP-seq with biotin ChIP-seq data, which shows much higher correlation with ENCODE CTCF ChIP-seq data.

Fig. 3a. Venn diagram showing the overlapping of Biotin-enriched CTCF or CTCF-s peaks with a normal CTCF ChIP-seq from ENCODE(GSE33213).

- Please describe the findings in the context of the crystal structures of CTCF ZFs and its contacts with DNA (Hashimoto et al, 2017 and Yin et al, 2017). In these studies, zinc fingers 1 and 2 do not make base specific contacts. It is unclear from the paper if ZF 3 is still intact or not in CTCF-s.

Reply: We have followed reviewer #3's suggestion and have cited the references suggested. We have emphasized that ZF3 is not intact in CTCF-s on page 4 as “***This putative shorter isoform (we termed CTCF-s) lacks the sequences encoding N-terminal domain plus 2.5 zinc fingers (ZFs), but still effectively contains 8 intact ZFs and a full length C-terminal domain (Fig. 1a).***” For characterizing CTCF-s specific base contacts, we have obtained much better ChIP-seq data for CTCF and CTCF-s using a biotin tag, and we have expanded the description of our findings in the context of the two papers as “***Recent crystal structural analysis of CTCF ZF1-11 have shown that the core 15 bp DNA binding motif was mainly specified by ZF3-7. As CTCF-s has lost ZF1-3, we were interested to know if the DNA consensus motif was also altered. De novo motif discovery indeed reported a 2 bp truncation in the consensus motif, which are the base pairs that Z1-3 makes contact with (Hashimoto et al, Mol Cell 66,711-720(2017); Yin et al., Cell Res 27, 1365-1377(2017)) (Fig. 3c).***”, showing that specific base contact for ZF3 were eliminated after losing the intact structure of ZF3.

- In fig 3D and E, In the EMSA when CTCF is in excess, the entire DNA shifts up. However, when CTCF-s is in excess, there is no reduction in the signal of free probe. What is the the probe that is used here – no information is provided? As per the ChIP-seq data, what is the efficiency of binding of CTCF and CTCF-s to the probe?

Reply: We used a 33 bp DNA probe from our CTCF ChIP-seq peaks, which contains canonical CTCF binding motif for our EMSA experiment. We have provided these oligo sequences in the **methods section** of the revised manuscript as **“DNA oligos were synthesized by Guangzhou IGE biotechnology. The forward probe sequence: 5’-(cy5)CCC, ATG, GCT, GGC, CAC, CAG, GGG, GCG, GCA, CAG, ACC-3’, the reverse probe sequence: 5’-GGT, CTG, TGC, CGC, CCC, CTG, GTG, GCC, AGC, CAT, GGG-3’.”**

As per the ChIP-seq data, biotin-CTCF binding signals were generally stronger than biotin-CTCF-s, suggesting that the efficiency of CTCF binding to DNA is strong than that of CTCF-s.

- Relative to figure 4: a co-IP of CTCF-s /CTCF and cohesin is required to test whether CTCF-s loses the ability to bind cohesin.

Reply: We performed this experiment in the previous manuscript. Specifically, we did FLAG tagged CTCF-s/CTCF co-IP experiments and blotted a cohesin subunit, RAD21 in **Supplementary Fig. 9d**. And we observed that the binding of CTCF-s to cohesin is much weaker than that of CTCF.

- In figure 4, Hi-ChIP should be performed with a FLAG antibody, comparing CTCF and CTCF-s overexpression. This will test whether CTCF-s on its own can make loops.

Reply: We have followed reviewer #3's suggestion and further performed HiChIP experiment with anti-FLAG antibody for both FLAG-CTCF and FLAG-CTCF-s. Our data indicated that a lot of CTCF loops were observed at chromosome 1 (**Fig. 1 for reviewer #3**, below), suggesting CTCF could successfully bridge loops. However, few CTCF-s loops were displayed at chromosome 1 (**Fig. 1 for reviewer #3**), indicating CTCF-s could not form CTCF-s specific loops or could only form very weak chromatin loops that could not be detected by FLAG HiChIP.

Fig.1 for reviewer #3. IGV diagrams showing FLAG-tagged normalized HiChIP tracks and their loops for the whole chr1. Top 5 tracks were two biological replicates (BR1 and BR2) of FLAG HiChIP tracks for FLAG-CTCF (purple), two biological replicates (BR1 and BR2) of FLAG HiChIP tracks for FLAG-CTCF-s (maroon) and one FLAG HiChIP tracks (grey) for FLAG alone as input.

- Overall, the connection between CTCF/Cohesin peak changes and gene expression changes need further investigation. This is the most interesting finding so the authors need to explore whether there is differential gene expression related to loss of CTCF and cohesin binding. For example is CTCF-s overexpression linked to changes in loops and gene expression.

Reply: We thank reviewer #3's for his or her insightful comments. In this revised manuscript, we have provided more results about the relationship between differential gene expression and loss of CTCF and cohesin binding after CTCF-s gain of function. Loss of CTCF and cohesin binding tends to

occur at promoter regions of down-regulated genes by CTCF-s gain of function (Fig. 5b,c). Of those up-regulated genes, CTCF binding at promoters did not alter upon CTCF-s gain of function, while the distance to the closest enhancers tends to be closer than the down-regulated genes (Fig 5d). This is consistent with a recent report in *Cell* by Nora et al. (2017).

Fig. 5. CTCF-s competition and transcriptional regulation. (a) Hierarchical clustering of differentially expressed (DE) gene profiles after FLAG-CTCF-s gain-of-function in HeLa-S3 cells. Fold-change is relative to the mean of the FLAG control. (b) Analysis of CTCF binding at Promoters of DE genes from a. (c) CTCF-s down-regulated genes tend to decrease CTCF binding at their promoter regions. (d) CTCF-s up-regulated genes tend to lie at shorter genomic distance to neighboring enhancers than down-regulated genes.

- Page 12: how do the authors explain that STAT1 phosphorylation was activated? They need to look at this in the context of chromatin bound phosphorylated STAT1 and a ChIP-seq of STAT1.

Reply: We followed the reviewer #3's suggestion and performed ChIP-seq for both STAT1 and phosphorylated STAT1. However, the STAT1 and pSTAT1 antibodies purchased from Cell Signaling Technology did not work effectively for ChIP experiments. Previous successful ChIP-seq data from other research groups used a pSTAT1 Santa Cruz Biotech antibody. However, the company no longer sells that antibody. Therefore, we have removed the STAT1 part from the manuscript in the revised version.

- Why does knocking down one gene, IFI6 impact apoptosis? Is this the only gene in the interferon pathway that regulates apoptosis?

Reply: IFI6 is not only the gene in the interferon pathway that regulates apoptosis as other genes have also been reported to regulate apoptosis (Stawowczyk et al., JBC, 286,7257-7266(2011)). However, from our RNA-seq and RT-qPCR results, we noticed that IFI6 is the most obvious gene activated by CTCF-s gain of function. Therefore, we focus on the role of IFI6 on apoptosis and found that knockdown of IFI6 could partially inhibit the activation of cell apoptosis triggered by CTCF-s gain of function.

- It is not clear how over-expression of CTCF-s leads to an inflammatory expression profile – is this true the other way round ie does inflammation (or treatment of cells with IFN α 2b) lead to over-expression of CTCF-s?

Reply: We treated *ex vivo* derived PBMC primary cells with varied concentrations of recombinant human IFN α 2b protein with bioactivity that could activate an immune response. We did not observe a significant increase of CTCF, but interestingly, we did observe increased expression of CTCF-s by IFN α 2b treatment in PBMC cells. Hence, inflammation does activate the expression of CTCF-s which is true in the other way around (**Fig. 2 for reviewer #3**).

Fig. 2 for reviewer #3. TaqMan RT-qPCR analysis of the relative expression levels of CTCF and CTCF-s in *ex vivo* derived PBMCs after treating with varied concentration of IFN α 2b.

- Relative to figure 6: the choice of the example would be more robust if loop changes match with CTCF/CTCF-s binding changes. From the figure 6d, the only peak that really looks different between CTCF and CTCF-s is not at a loop

anchor. The authors should show IGV 1-dimensional track of the Hi-ChIP to compare with ChIP-seq peaks.

Reply: We have included a 1-dimensional track of CTCF HiChIP data in the revised manuscript. As the varied strength of CTCF binding at different genomic binding sites, some weak CTCF binding sites could not be observed at the scale we presented. However, we also marked those weak CTCF binding sites and the binding direction by red arrows. In the revised manuscript, we could observe that all of those loop anchors contain CTCF binding sites. Though some of the differential CTCF binding sites could not form chromatin loops, other binding sites (3rd, 8th and 10th grey marked anchors) mediated by CTCF could form chromatin loops.

Fig. 6i. (i) CTCF HiChIP interactions at *IFI6* locus displayed with the CTCF ChIP-seq and 1-dimensional track of CTCF HiChIP in FLAG control and FLAG-CTCF-s overexpressed cells. The directionality of the motifs at CTCF loop anchors associated with peaks was indicated with red arrows and gray bars.

- It is unclear how well the IP part of the Hi-ChIP experiment worked. Screenshots of Juicebox showing loops should be included, not only the arcs.

Reply: We have provided screenshots of Juicebox (at 500-kb, 25-kb and 10-kb resolution) showing loops of CTCF HiChIP in FLAG-overexpressed cells and in FLAG-CTCF-s-overexpressed cells in the previous manuscript (Fig. 4d). In the manuscript, we described Fig. 4d as “**We then examined the dynamics of chromatin looping mediated by CTCF upon CTCF-s gain-of-function. We firstly inspected the raw interaction matrix at progressively higher resolutions, and found that chromatin features were similar to Hi-C interaction map at 500-kb, 25-kb and 10-kb resolution (Fig. 4d).**”.

- A screenshot of the raw interaction map at specific loci with differential loops should be shown in supplemental data to support the author's claims that the loops are differential.

Reply: We have included a screenshot of the raw interaction map at specific loci (*MDM2/CPM* loci and *IFI6* loci) with differential loops in the Supplementary **Fig. 10c and 10d**.

Supplementary Fig. 10c and d. (c) Screenshot of raw interaction map at *MDM2/CPM* loci from FLAG control and FLAG-CTCF-s overexpressing HeLa-S3 cells. Numbers below the interaction maps correspond to maximum signal in the matrix. (d) Screenshot of raw interaction map at *IFI6* loci from FLAG control and FLAG-CTCF-s overexpressing HeLa-S3 cells. Numbers below the interaction maps correspond to maximum signal in the matrix.

- 4C-seq should be performed from different viewpoints across the region to validate their claim. 3C is very biased and does not give a full picture of interactions across the region. Furthermore, it appears that there are no controls for the 3C experiment shown.

Reply: We are appreciated the reviewer #3's suggestion. Actually, we have inquired many companies to get the BAC clone for the 3C control, none of them provided the BAC clones for the *IFI6* locus, thus we could not provide the BAC control for the 3C experiment. Instead, we used genomic GAPDH locus as control. To address the reviewer's concern, we have tried our best to provide 4C-seq results instead. During the last three months, we performed 4C-seq experiments with fourteen 4C primer sets, different PCR amplification condition, performed high-output sequencing five times, however none of them gave us enough data around this region for 4C-seq data analysis. Another

problem we encountered was that majority of the sequencing reads were non-specific PCR products, which means valid read depth was low. Due to the limit of enzyme selection for 1st and 2nd enzymatic digestion for the promoter anchor region, the regions for primer design were also limited. We proposed that this locus is less sensitive and difficult for 4C-seq experiments. We would like to highlight that the use of *Actin* or *GAPDH* instead of a BAC clone as control is often used in the literature, and in our opinion is a suitable control (eg: Sciacovelli, et al., *Nature* **537**, 544-547(2016), Elisa Barbier et al., *Mol Cell* **71**, 103-116(2017), Gao P. et al., *Cell* **174**, 576-589(2018)).

- Overall, the discussion is superficial. There are major points not addressed in the paper that should be addressed in the conclusion as a perspective of future directions.

Reply: We have accepted reviewer #3's criticism and have revised our discussion and provided perspective of future directions about our main discovery in the revised manuscript.

We have discussed balance of control of RBPs on gene alternative splicing and cell fate determination and added the following sentences in the revised manuscript. ***"In this study, we identified an alternatively spliced CTCF-s short isoform and identified 9 RBP candidates that positively or negatively control CTCF alternative splicing through screening. The balance of these splicing regulators precisely controls the splicing ratios of specific genes in the living compartments (Fu & Ares, Nat Rev Genet 15,689-701(2014)), and imbalance of splicing factors may contribute to biological and/or developmental abnormality or carcinogenesis (Dvinge et al., Nat Rev Can 16, 413-430(2016)). Therefore, the properly regulation of CTCF alternative splicing by these 9 newly identified splicing factors could be important for CTCF functions in regulating gene expression and cell fate determination."***

We have also discussed DNA binding motif and competition between CTCF and CTCF-s and added the following sentences into the discussion ***"Two recent investigations about the crystal structure of CTCF ZFs specify nucleotides bound by each ZFs (Hashimoto et al, Mol Cell***

66,711-720(2017); Yin et al., *Cell Res* 27, 1365-1377(2017)). Consistently, biotin-CTCF specifically binds the 15 bp core DNA motif, and biotin-CTCF-s, which lacks 7 aa (HKCPDCD) of the 24-aa ZF3, no longer recognizes the 2 core motif (C/G T/C) for ZF3 and only reserves G/A with decreased specificity. Our data suggested that CTCF-s competes with CTCF to reduce cohesin binding, and further disrupt chromatin looping mediated by CTCF, providing a new eviction mechanism for CTCF-mediated chromatin looping that is not triggered by mutations of CTCF binding sites or alteration of CTCF binding orientation (Tang et al., *Cell* 163,1611-1627(2015); Guo et al., *Cell* 162,900-910(2015)). It was suggested that recruitment of cohesin to most CTCF binding sites is in a CTCF-dependent manner (Parelho et al., *Cell* 132,422-433(2008); Wendt et al., *Nature* 451,796-801(2008)). Consequently, the reduction of cohesin could be explained if CTCF-s is impaired to recruit CTCF/cohesin, and so competes away CTCF and cohesin. Indeed, the interaction of CTCF and cohesin seems much stronger than that of CTCF-s and cohesin (Supplementary Fig. 9d), which might be another reason to characterize the decreases of cohesin- and CTCF-binding in their common targets and to explain why CTCF-s gain-of-function changes CTCF-mediated chromatin looping in the genome.”

We have also added CTCF-s specific binding, CTCF splicing pattern and future perspectives into the discussion as below. “*In this study we have mainly focused on the roles of CTCF-s in the context of canonical CTCF binding, however, there was 12,644 CTCF-s binding sites that did not overlap with CTCF peaks (Fig. 3a). This intriguing observation suggests that in addition to its competitive function, CTCF-s binds to genomic regions independently of CTCF and provides an additional layer of genome organization. Further, the splicing pattern of CTCF varies in different cell types, providing a diversity of CTCF-mediated higher-order chromatin organization possibilities in different cell types. Therefore, the investigation of CTCF alternative splicing mechanisms and characterization of the biological or pathological functions, especially in cancer, of this newly reported CTCF short isoform will*

need to be addressed in the future.”

Finally, we have summarized this study as below. ***“In summary, we conclude that ectopic expression of CTCF-s can disrupt the chromatin architecture maintained by CTCF. Therefore, CTCF-s may act as a modulator or ‘fine-tuner’ of CTCF activity, balancing CTCF-mediated chromatin organization and transcriptional regulation. The competition between CTCF and CTCF-s ultimately leads to an effect on cell phenotype as the cells activate an apoptotic pathway. Our study demonstrates the importance of alternatively spliced isoforms in chromatin and cellular activity.”***

We provided a clearer statement of the novelty and significance of this study and a spotlight for further investigation and we hope that this discussion could meet the reviewer’s expectation and readers’ interests.

Minor Comments

- The culture conditions of all cell lines should be described in the Method section.

Reply: We have added culture conditions of all cell lines in the revised method section. ***“HeLa-S3, 293T, GZF2, IRM90, MCF7, HepG2, HCT116, SHSY5Y cells were cultured in DMEM/High Glucose (Hyclone) supplemented with 10% FBS. Kasumi-3 cells were cultured in DMEM/High Glucose (Hyclone) supplemented with 20% FBS. K562, A549, COLO 829 and PBMC cells were cultured in 1640 medium (Hyclone) supplemented with 10% FBS. HPTC cells were cultured in DMEM/F12 medium (Hyclone) supplemented with 10% FBS, 0.1 mM NEAA, 1 mM L-Glutamax, 0.1 mM β -mercaptoethanol and SingleQuot Kit CC-4127 REGM (Lonza). Human ESC cell lines H1 (Wi Cell), H9 (Wi Cell), HN10 and uiPS cells were cultured in mTeSR1 medium (STEMCELL Technologies) on matrigel (Corning)-coated plates. NSP cells were cultured in N2B27 medium of a 1:1 mixture of DMEM/F12 and Neurobasal medium supplemented with 1 \times NEAA, 1% N2 (Invitrogen), 2% B27 (Invitrogen), 5 μ g/ml insulin, 2 μ g/ml heparin, 100 μ M β -mercaptoethanol, 5 μ M SB431542 and 5 μ M***

Dorsomorphin. Cultured cells were maintained at 37°C in a 5% CO₂ incubator.

- All venn diagrams should be true to scale. For eg, in Supp fig. 7, 30817 seems bigger than 48822 by eye.

Reply: We have followed the reviewer #3's suggestion and scaled all the Venn diagrams in the figures as suggested.

- The level of expression with respect to the transgene should be shown.

Reply: We have followed reviewer #3's suggestion and included the level of expression with respect to the transgene. We performed RT-PCR for the N terminal (CTCF-N), specifically targeting to CTCF long isoform and C terminal (CTCF-C) of CTCF targeting to both long and short isoforms of CTCF by using different set of primers.

- Page 5: more information is required about the in silico prediction of splice sites.

Reply: We have followed reviewer#3's suggestion and added more detailed information about the in silico prediction of splice sites in the Method part as below. ***“We have taken advantage of the website from Christopher Burge’s laboratory, it provided detailed tips for in silico prediction of splice sites. For scoring the 5’ splicing site (http://genes.mit.edu/burgelab/maxent/Xmaxentscan_scoreseq.html), the sequence must be 9 bases long, which contains 3 bases in exon and 6 bases in intron. And for scoring the 3’ splicing sites (http://genes.mit.edu/burgelab/maxent/Xmaxentscan_scoreseq_acc.html), the sequence must be 23 bases long, which contains 20 bases in the intron and 3 bases in the exon. Based on these rules, we have used CTCF genomic sequence for in silico prediction of these splice sites.”***

- Supplementary figure 6d: the molecular weight is not shown. How do the authors explain that CTCF-s is more abundant than CTCF? What are all the bands for CTCF? The text in page 7 says that CTCF and CTCF-s are mainly

located in the nucleus, but there is still a lot in the cytoplasm. For all Westerns the nucleus should be divided into 2 fractions: nucleoplasmic and chromatin-bound.

Reply: We have added the molecular weight for this panel in the revised manuscript. For this panel, we used FLAG-tagged overexpressed CTCF and CTCF-s, therefore, as shown in **Supplementary Fig. 6a**, the overexpression of CTCF-s is much stronger than overexpression of CTCF. We have marked the bands for both CTCF and CTCF-s in the figures and propose that other bands might be CTCF degraded products as none of these bands are shown in the detection of CTCF-s (FLAG). In combination with **Supplementary Fig. 6c** and **Supplementary Fig. 6d**, although a lot of CTCF-s is located in the cytoplasm, it is more abundant in the nucleus, especially so for CTCF. Following reviewer #3's suggestion, we further divided the nucleus fraction into nucleoplasmic and chromatin-bound fractions (see revised method), and found that both CTCF and CTCF-s locate in the chromatin-bound fraction.

- **Figure 3 and figure 4: it is not clear what the control situation is. Cluster 2 should be shown in figure 3g as a comparison.**

Reply: The control here means FLAG control. Comparison of cluster 2 was originally shown in Supplementary Fig. 7c. In the revised manuscript, we have shown this panel in **Fig. 3g**.

- **Figure 4e: how many total CTCF and CTCF-s loops?**

Reply: For **Fig. 4e**, we only performed CTCF HiChIP and determined the dynamics of CTCF loops upon CTCF-s overexpression, therefore, all these loops in **Fig. 4e** were CTCF loops. For CTCF HiChIP experiments in FLAG control cells, there were 239,440 CTCF loops, after processing these CTCF loops with $FDR < 0.05$ by mango. For CTCF HiChIP experiment in FLAG-CTCF-s overexpressed cells, there were 137,944 CTCF loops remained after filtering these CTCF loops with $FDR < 0.05$ by mango. We include the loop information in **Supplementary Table 5**.

- Figure 4i: what is the directionality of the motifs at CTCF loop anchors associated with peaks?

Reply: We apologize for not showing these clearly. In this revised manuscript, we have added the directionality of the motifs at CTCF loop anchors that were associated with peaks in **Fig. 4i**. In addition, we also provided the directionality of the motifs at CTCF loop anchors in **Fig. 6i**.

- Figure 5g: Does it mean that expressing CTCF exogenously leads to survival of cells over and above that of WT conditions? This doesn't really make sense.

Reply: In a previous study, we have reported that overexpression of full length CTCF (POZN-overexpression system) leads to the augmentation of cell sizes and cell growth in HeLa S3 cells (Huang et al., JBC, 288, 26067-26077(2013)). In this study, we could get the same results by using pSin-FLAG vector after CTCF gain-of-function. And our data indeed shows that expressing CTCF exogenously leads to increased cell growth in HeLa-S3 cells.

- Page 12: what is the read depth for all the Hi-ChIP samples? A summary table of valid pairs of interactions and loops should be included containing all the information before and after any cutoff and processing step.

Reply: We sequenced about 180 M - 200 M reads for each HiChIP replicate. For data analysis, we combined the two biological replicates and processed the data according to the data processing protocol (see method section). Loops with a distance between 5kb~2Mb were used as primary CTCF loops, and those loops with false discovery rate (FDR) <0.05 were used for the downstream analysis. In the revised manuscript, we have provided the valid pairs of interactions and loops before and after our data processing steps in **Supplementary Table 5**.

- Page 32: for analysis of differential Hi-ChIP, please explain why peaks were extended 500 bp and what the authors mean by restriction fragments containing the merged peaks that were used as loop anchors. Please include a schematic representation of how the analysis is performed.

Reply: Hichipper was developed by Caleb A Lareau and Martin J Aryee (Lareau CA, Aryee MJ. *Nature methods* **15**, 155-156 (2018)). The authors provided the files for the hichipper analysis in the websites:

<https://hichipper.readthedocs.io/en/latest/content/Configuration.html#restriction-fragment-aware-padding>. As noted in orange, defined peaks are automatically padded by some integer width from the --peak-pad flag. By default, this pad extends 500 bp in either direction. Padding the peaks boosts the number of PETs that can be mapped to loops (Fig. 3 for reviewer #3).

For schematic representation of how the analysis is performed, we strictly follow the pipeline from the Nature Method paper, and we provided a snapshot from this paper below (Fig. 4 for reviewer #3).

Fig. 3 for reviewer #3. Parameter explanations.

Fig. 4 for reviewer #3. Snapshot of the hichipper analysis pipeline from Nat. Methods, 2018.

- For the differential Hi-ChIP loops, does the p-value cutoff correspond to the raw or adjusted p-value.

Reply: For the differential HiChIP loops, we used raw p-value during diffloop step, and then filtered out loops using replicates which were added in the revised method section.

- In Supplementary figure 6a, do the primers detect ectopic CTCF as well as the endogenous CTCF? A schematic representation of the primer location should be included. For CTCF, the level of CTCF-N and CTCF-C should be comparable as the primers for the full length CTCF have both ends of the protein.

Reply: We agreed with the third reviewer about this question. Both CTCF-N and CTCF-C can detect ectopic CTCF as well as the endogenous CTCF. As suggested, we have included a schematic representation of the primer locations for **Supplementary Fig. 6a** in this revised manuscript. Theoretically, the level of CTCF-N and CTCF-C should be comparable, however, the actual amplification efficiency for CTCF-N and CTCF-C were not comparable.

- In the method section, the references of the antibodies used for ChIP and Hi-ChIP should appear.

Reply: We specify the antibodies used in this study in the revised manuscript and added the detailed information into **Supplementary Table 3**.

Reviewers' comments:

Reviewer #1 (Remarks to the Author):

As for the original submission, I restrict my comments to the splicing aspects. The existence of the CTCFs isoform is not controversial – as stated it is already an annotated isoform. The new western blot data (Fig 1f) indicate that it is usually a minor isoform, although it can represent as much as 40% of total CTCF. The qRT-PCR and western blot data also more strongly support the specificity of the shRNAs (Fig 1g, h). This provides a sound basis for the remaining work on the potential function of CTCFs.

I still have concerns with the experiments that start to address the mechanism of exon 3 and 4 alternative splicing (Fig 2 and associated supplementary data). My advice would be to remove Fig 2, the associated supplementary Figs S2 and S3, and the text on p5-7. The remainder of the manuscript, which addresses a more coherent set of questions, could be judged on its own merits.

Specific comments

The rationale for the experiments and the hypotheses being tested in Fig 2 and Supp Figs 2 and 3 are not well articulated. Typically, investigations of alternative splicing address how it is regulated between different cell types or conditions where the splicing pattern (percent inclusion) varies substantially. The experiments here all seem to be carried out in 293T cells, where exons 3 and 4 appear to be nearly fully included. In my view, these experiments are not necessary for this manuscript, and actually represent a weak point in the paper.

The experiments using RNA pulldown are premature without first having carried out any sort of mutagenesis to identify regulatory elements that are important for exon 3 and 4 inclusion or skipping. Supp Fig 2c and d show a minigene reporter system that could be used as a starting point for such an analysis (using RT-PCR which can give a quantitative readout rather than GFP fluorescence). Any biotinylated RNA will pull down RNA binding proteins from a cellular extract, and a probe that encompasses splice sites will naturally tend to pull down some splicing factors. These sorts of experiments are therefore more informative when a mutant sequence is available and differential binding can be monitored that correlates with altered activity.

The data in Fig 2g shows some statistically significant alterations in the ratio of CTCFs and CTCF in response to knockdown of various RNA binding proteins. But as I originally pointed out, the data are normalized in such a way that it is impossible work out the actual magnitude of change in splicing. For mechanistic analyses of splicing regulation it is not sufficient to determine the fold-change in expression of individual isoforms. A ratiometric approach is needed that can determine percent spliced in (PSI) – or occasionally investigators quantify the ratio of splicing products. The simplest approach is to use RT-PCR that can detect both isoforms (as in Supp Fig 2d). We are not told (as far as I could see) the cell type in which the knockdowns were carried out in Fig 2g. If it is 293T the data in Fig 1 (and Supp 2d) suggest that the PSI for exons 3 and 4 is 90%. This is then normalized to a “Relative CTCF/CTCFs ratio” of 1.0 in Fig 2g. In the knockdowns, the largest change in ratio from 1 to 3 (sh hnRNPC #2) transforms to a change in PSI of ~90 to 77%. This may be a statistically significant change, as indicated, but it is far from a substantial effect and does not provide grounds to support the claim that “these 9 RBPs are critical regulators of CTCF alternative splicing” (page 7, line 16).

In sum these data show that exons 3 and 4 are partially skipped in a minigene context, that various RNA binding proteins bind to parts of the transcript, and that knockdown of some of these RNA binding proteins can modestly affect the splicing pattern.

Reviewer #2 (Remarks to the Author):

The authors did perform an extensive work in order to strengthen their study on the role of the alternative form of CTCF and have, in my opinion, addressed most of the concerns raised during the first round of revision. Although we can still see the two parts of the study, as mentioned in the first review, and some questions are still open, the article has been clearly improved. As mentioned in the previous review, this study brings interesting findings for the fields of gene regulation and genome architecture.

Reviewer #3 (Remarks to the Author):

The authors have made significant improvements to the manuscript and addressed all our concerns. I would be happy to recommend publication in Nature Communications.

Rebuttal Letter

Reviewers' comments:

Reviewer #1 (Remarks to the Author):

As for the original submission, I restrict my comments to the splicing aspects. The existence of the CTCFs isoform is not controversial – as stated it is already an annotated isoform. The new western blot data (Fig 1f) indicate that it is usually a minor isoform, although it can represent as much as 40% of total CTCF. The qRT-PCR and western blot data also more strongly support the specificity of the shRNAs (Fig 1g, h). This provides a sound basis for the remaining work on the potential function of CTCFs.

I still have concerns with the experiments that start to address the mechanism of exon 3 and 4 alternative splicing (Fig 2 and associated supplementary data). My advice would be to remove Fig 2, the associated supplementary Figs S2 and S3, and the text on p5-7. The remainder of the manuscript, which addresses a more coherent set of questions, could be judged on its own merits.

Response: By following the reviewer#1's suggestion, we have removed original Figure 2, the associated supplementary Figures S2, S3, S4 and S5 as well as the text on pages 5-7.

Specific comments

The rationale for the experiments and the hypotheses being tested in Fig 2 and Supp Figs 2 and 3 are not well articulated. Typically, investigations of alternative splicing address how it is regulated between different cell types or conditions where the splicing pattern (percent inclusion) varies substantially. The experiments here all seem to be carried out in 293T cells, where exons 3 and 4 appear to be nearly fully included. In my view, these experiments are not necessary for this manuscript, and actually represent a weak point in the paper.

Response: We are appreciated for the reviewer#1's helpful comments. We have removed these data from the manuscript as suggested.

The experiments using RNA pulldown are premature without first having carried out any sort of mutagenesis to identify regulatory elements that are important for exon 3 and 4 inclusion or skipping. Supp Fig 2c and d show a minigene reporter system that could be used as a starting point for such an analysis (using RT-PCR which can give a quantitative readout rather than GFP fluorescence). Any biotinylated RNA will pull down RNA binding proteins from a cellular extract, and a probe that encompasses splice sites will naturally tend to pull down some splicing factors. These sorts of experiments are therefore more informative when a mutant sequence is available and differential binding can be monitored that correlates with altered activity.

Response: We thank the reviewer#1's for his/her grateful comments. By following the reviewer#1's suggestions, we have removed these data from the current version of manuscript, but we will perform additional mutant experiments to expand this part of work for another story in the future.

The data in Fig 2g shows some statistically significant alterations in the ratio of CTCFs and CTCF in response to knockdown of various RNA binding proteins. But as I originally pointed out, the data are normalized in such a way that it is impossible work out the actual magnitude of change in splicing. For mechanistic analyses of splicing regulation it is not sufficient to determine the fold-change in expression of individual isoforms. A ratiometric approach is needed that can determine percent spliced in (PSI) – or occasionally investigators quantify the ratio of splicing products. The simplest approach is to use RT-PCR that can detect both isoforms (as in Supp Fig 2d). We are not told (as far as I could see) the cell type in which the knockdowns were carried out in Fig 2g. If it is 293T the data in Fig 1 (and Supp 2d) suggest that the PSI for exons 3 and 4 is 90%. This is then normalized to a “Relative CTCF/CTCFs ratio” of 1.0 in Fig 2g. In the knockdowns, the largest change in ratio from 1 to 3 (sh hnRNP #2) transforms to a change in PSI of ~90 to 77%. This may be a statistically significant change, as indicated, but it is far from a substantial effect and does not provide grounds to support the claim that “these 9 RBPs are critical regulators of CTCF alternative splicing” (page 7, line 16).

Response: We thank the reviewer#1 for his/her professional advice. By following the reviewer#1's suggestions, we have removed these data from the current version of manuscript, but we will perform PSI assay to study CTCF alternative splicing as another story.

In sum these data show that exons 3 and 4 are partially skipped in a minigene context, that various RNA binding proteins bind to parts of the transcript, and that knockdown of some of these RNA binding proteins can modestly affect the splicing pattern.

Response: We agree with the reviewer #1's comments. We have followed the reviewer#1's comments and deleted this part of work from the current manuscript, but we will strengthen this part work later as suggested.